# DASB - Discrete Audio and Speech Benchmark

## Abstract

Discrete audio tokens have recently gained considerable attention for their potential to connect audio and language processing, enabling the creation of modern multimodal large language models. Ideal audio tokens must effectively preserve phonetic and semantic content along with paralinguistic information, speaker identity, and other details. While several types of audio tokens have been recently proposed, identifying the optimal tokenizer for various tasks is challenging due to the inconsistent evaluation settings in existing studies. To address this gap, we release the Discrete Audio and Speech Benchmark (DASB), a comprehensive leaderboard for benchmarking discrete audio tokens across a wide range of discriminative tasks, including speech recognition, speaker identification and verification, emotion recognition, keyword spotting, intent classification, event sound detection, and music genre classification as well as generative tasks such as speech enhancement, separation, and text-to-speech. Our results show that, on average, semantic tokens outperform compression tokens across most discriminative and generative tasks. However, the performance gap between semantic tokens and standard continuous representations remains substantial, highlighting the need for further research in this field.

## 1 Introduction

Traditional speech and audio processing systems have long relied on handcrafted low-level features such as *Mel-Frequency Cepstral Coefficients* and *Filterbanks* Rabiner & Juang (1993). Recently, self-supervised learning (SSL) led to outstanding performance improvements by learning more complex, robust, and general speech features through deep neural networks. Notable models include Wav2Vec2 Baevski et al. (2020), WavLM Chen et al. (2022), and HuBERT Hsu et al. (2021). In all these cases, the rich information in speech and audio signals is encoded into a sequence of continuous vectors. Even though continuous vectors have proven effective in capturing the complex details embedded in speech and audio, there is a growing interest in discrete representations. Discrete audio representations, known as audio tokens, transform the original waveform into a finite set of vectors. These tokens are derived using methods such as quantization of self-supervised learning (SSL) models Polyak et al. (2021); Wells et al. (2022); Chung et al. (2021), neural compression techniques (codecs)Zhang et al. (2024); Du et al. (2023), or hybrid approaches Zhang et al. (2024); Du et al. (2023) that combine both methods.

*What is driving the interest in audio tokens?* Arguably, this trend is linked to the remarkable success of autoregressive Large Language Models (LLMs) such as LLama Touvron et al. (2023), PALM Chowdhery et al. (2024), BERT Devlin et al. (2019), and GPT Liu et al. (2023). Unlike audio, these models operate on text, which is inherently discrete. Inspired by their effectiveness, researchers are exploring audio language models van den Oord et al. (2017); Rubenstein et al. (2023); Wang et al. (2023c;d;a;e); Agostinelli et al. (2023); Kreuk et al. (2023) by representing the audio as a sequence of discrete tokens. Moreover, audio and text tokens can be naturally combined, paving the way for the development of modern multi-modal LLMs Gemini Team, Google (2023) capable of processing audio, text, and visual data. Discrete tokens also simplify audio generation tasks like speech enhancement and synthesis by turning them into classification problems instead of regression models Goodfellow et al. (2016). Finally, they also enable efficient data compression for better transmission and storage. The main drawback of audio tokens is the inevitable loss of information introduced by the discretization process. We ideally aim for audio tokens that preserves crucial information of the original waveform, including phonetic and linguistic content, speaker identities,

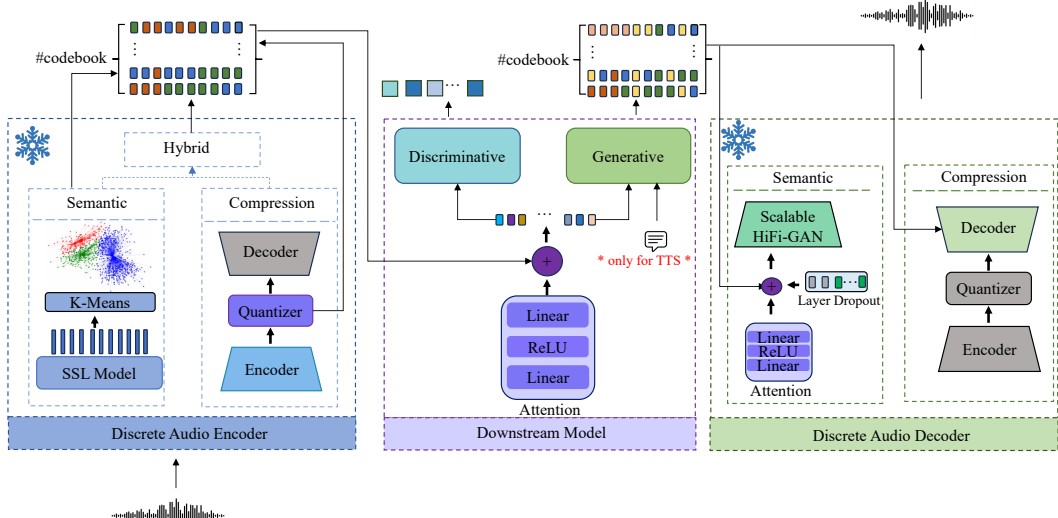

Figure 1: The workflow of DASB consists of three steps. First, a discrete audio encoder converts the audio signal into discrete tokens (*left*). Then, the tokens are combined using attention and fed to a neural model for the final prediction (*middle*). For generative tasks, the predicted tokens are passed to a discrete decoder, which converts them back into an audio waveform (*right*). Both the encoder and decoder are pretrained and frozen during downstream model training.

emotions, and other paralinguistic cues. However, despite the growing trend toward audio tokens, there is still a lack of standardized evaluation benchmarks, with different studies employing varied experimental settings Wu et al. (2024); Puvvada et al. (2024); Wang et al. (2024); Zhang et al. (2023); Chang et al. (2024). Without a consistent framework for measuring and comparing performance, it becomes challenging to determine which audio tokens perform optimally across various tasks.

To address this gap, we introduce the **D**iscrete **A**udio and **S**peech **B**enchmark (DASB). DASB systematically assesses various audio tokens across several common speech processing tasks. In particular, our contribution is the following:

- We benchmark a diverse set of discrete audio encoders from all three categories: semantic (Discrete HuBERT, Discrete WavLM, Discrete Wav2Vec2), *compression* (EnCodec Défossez et al. (2023), DAC Kumar et al. (2023)), and *hybrid* (SpeechTokenizer Zhang et al. (2024)).
- We assess a broad set of discriminative tasks, including speech, speaker, and emotion recognition, keyword spotting, and intent classification. Moreover, we extend our benchmark to sound and music tasks, such as event sound detection and music genre classification. For generative tasks, we evaluate speech enhancement, separation, and text-to-speech. For a more reliable assessment, we consider different downstream architectures for each task, following the insights in Zaiem et al. (2023). To the best of our knowledge, this is the first comprehensive benchmark of audio tokens that covers both discriminative and generative tasks.
- We publicly release DASB[1] as a modular code repository.

## 2 RELATED WORK

Several research efforts have recently explored using discrete audio tokens as an alternative to continuous features. Some studies focused on using discrete features for speech recognition and speech translation Chang et al. (2024); Zhang et al. (2023); Chang et al. (2023), specifically evaluating the tokens obtained from the quantized versions of the HuBERT model. Similarly, Yang et al. Yang et al. (2024) examined discrete features for speech recognition and text-to-speech. Audio tokens

---

[1]We will release the code and pretrained model for camera-ready verison.

have been proposed for speech enhancement as well. For example, Wang et al. Wang et al. (2024) investigated the application of semantic tokens to speech enhancement, whereas Erdogan et al. Erdogan et al. (2023) proposed a hybrid tokenizer called TokenSplit for both speech enhancement and separation.

While previous studies investigated the use of compression or hybrid tokens Wang & Székely (2024); Wang et al. (2023b); Kharitonov et al. (2023), these efforts were often limited to specific applications and a few audio tokenizers. In particular, previous benchmarking attempts focused on one category of tokenizers, either semantic or compression-based, and mostly on discriminative or generative tasks. For instance, Puvvada et al. Puvvada et al. (2024) compared the performance of EnCodec and DAC Kumar et al. (2023) for speaker verification, speaker diarization, and speech recognition. Mousavi et al. Mousavi et al. (2024) benchmarked various discriminative and generative tasks with semantic tokens. Wu et al. Wu et al. (2024) provided a comprehensive study of the quality of resynthesized sound with compression and hybrid tokenizers. The latter attempt used pretrained models for speech, speaker, and emotion recognition, and assessed how much information is preserved by feeding them resynthesized audio. However, it did not address the direct use of tokenized input for training downstream tasks, nor did it deeply analyze the role of semantic tokens. Our analyses, instead, suggest that semantic tokens outperform other tokenizers.

To the best of our knowledge, the proposed DASB benchmark is the first to compare several audio tokenizers from three categories (semantic, compression, and hybrid) across many discriminative and generative speech tasks of broad practical interests. Moreover, unlike previous works on discrete audio tokens, we draw inspiration from the findings presented in Zaiem et al. (2023) for reliably benchmarking continuous SSL representations and we consider different downstream architectures for each task. Similar to the approach taken by SUPERB wen Yang et al. (2021) for continuous representation, we offer a standardized evaluation benchmark where researchers can easily evaluate novel audio tokens.

## 3 BENCHMARK DESIGN

The pipeline of DASB, illustrated in Fig. 1, consists of three components: Audio Encoder, Downstream Model, and Audio Decoder. The main features of the considered tokenizers are summarized in Table 1, while Figure 2 reports the time and memory resources required by both encoders and decoders. The following subsections describe each module in detail.

### 3.1 DISCRETE AUDIO ENCODER

The audio encoder converts the audio signal into a sequence of discrete tokens. It is pretrained on large amounts of unlabeled data and remains frozen during the training of downstream tasks. Different encoders may compress the information in the original waveform at different rates. The compression level is measured by the bitrate, defined as:

$$\text{bitrate} = \log_2 V \cdot C \cdot R, \tag{1}$$

where $C$ is the number of codebooks, $V$ is the number of vectors in each codebook (vocabulary), and $R$ is the rate of codes per second. It is worth mentioning that a single sequence of tokens might be insufficient to capture the rich and complex information embedded in speech signals. The encoders thus often output multiple discrete sequences, with each sequence corresponding to a different codebook $C$. The encoders can operate at different bitrates simply by adjusting the number of codebooks $C$. For a fairer comparison, we define three different distinct bitrate ranges that we have identified from the literature Valin et al. (2012); Dietz et al. (2015); Zeghidour et al. (2021): *low* (0-1.5 kbps), *medium* (2.9-6 kbps), and *high* (24 kbps). We consider this approach to prevent the trivial conclusion that some audio tokens perform better than others simply due to a higher bitrate.

The design of DASB is flexible, allowing for easy integration and benchmarking of various tokenizers. Using the terminology from Borsos et al. (2023); Zhang et al. (2024), we categorize audio tokens into three classes: semantic, compression, and hybrid tokenizers. **Semantic** tokens Polyak et al. (2021); Wells et al. (2022); Chung et al. (2021) are generated by clustering or quantizing layers from SSL models Baevski et al. (2020); Chen et al. (2022); Hsu et al. (2021). The tokenization process typically involves selecting specific layers from a pretrained SSL model and applying the k-means

Table 1: Key Features of the Discrete Audio Encoders. #Params is computed for medium bitrate.

| Model | #Params | Sampling Rate | Bitrate (kbps) | | | #Codebooks | | |
|---|---|---|---|---|---|---|---|---|
| | | | low | medium | high | low | medium | high |
| Discrete HuBERT | 309.0M | 16KHz | 0.98 | 2.9 | - | 2 | 6 | - |
| Discrete WavLM | 309.0M | 16KHz | 0.98 | 2.9 | - | 2 | 6 | - |
| Discrete Wav2Vec2 | 309.0M | 16KHz | 0.98 | 2.9 | - | 2 | 6 | - |
| EnCodec Défossez et al. (2023) | 17.9M | 24KHz | 1.5 | 6.0 | 24.0 | 2 | 8 | 32 |
| DAC Kumar et al. (2023) | 22.4M | 24KHz | 1.5 | 6.0 | 24.0 | 2 | 8 | 32 |
| SpeechTokenizer Zhang et al. (2024) | 85.3M | 16KHz | 1.0 | 4.0 | - | 2 | 8 | - |

algorithm to group their representations. Semantic tokens primarily capture high-level information, such as phonetic, semantic, and syntactic information. They are not optimized for waveform reconstruction, making them potentially better suited for discriminative tasks like speech recognition. Recent studies, however, have shown that semantic tokens can also be effective in generative tasks Wang et al. (2024); Yang et al. (2024); Mousavi et al. (2024). We adopt the tokenization algorithm proposed in Mousavi et al. (2024). In particular, we consider three widely-used open-source SSL models: Wav2Vec2-large, WavLM-large, and HuBERT-large, each composed of 24 layers. Then, we cluster six of these layers using the k-means algorithm and select two layers from the lower part (1, 3) to capture low-level information, two from the middle layers (7, 12), and two from the higher layers (18, 23) to encode content and meaning as well.

**Compression** tokens Zeghidour et al. (2021); Défossez et al. (2023); Kumar et al. (2023) are mainly used for audio compression. They are trained to accurately reconstruct the original audio, making them potentially suitable for audio generation tasks. We integrated two publicly available compression-based tokenizers in our baseline. EnCodec Défossez et al. (2023) has three main components: (i) an encoder network $E$ consisting of a 1D convolution followed by a two-layer LSTM that processes the audio input and produces a latent representation $z$; (ii) a quantization layer $Q$ that compresses $z$ into $z_q$ using Residual Vector Quantization (RVQ) Zeghidour et al. (2021), where distinct codebooks quantizes residuals in multiple steps; and (iii) a decoder network $G$ that mirrors the encoder and reconstructs the time-domain signal $\hat{x}$ from $z_q$. The system is trained end-to-end to minimize reconstruction loss over time and frequency domains. It also adopts a perceptual loss using discriminators at different resolutions. EnCodec offers multiple models at low to medium bitrates (1.5 to 24 kbps). DAC Kumar et al. (2023) is an improved version of EnCodec. It combines advances in high-fidelity audio generation with better vector quantization techniques from the image domain, along with improved adversarial and reconstruction losses. DAC also supports quantizer dropout, allowing a single model to support variable bitrates.

**Hybrid** tokenizers Zhang et al. (2024); Du et al. (2023) unify semantic and acoustic tokens by disentangling different aspects of speech information hierarchically. SpeechTokenizer Zhang et al. (2024) is a unified speech tokenizer for large language models. It combines semantic and acoustic tokens, separating different speech information across RVQ layers. The model is based on RVQ-GANs, similar to EnCodec, and uses a convolutional encoder-decoder network from EnCodec. A two-layer BiLSTM replaces the original two-layer LSTM to improve semantic modeling. A semantic teacher guides the first RVQ quantizer, allowing the first layer tokens to capture content information effectively. With a residual structure, the subsequent quantizers capture the remaining paralinguistic information. SpeechTokenizer employs HuBERT as the semantic teacher to capture content information. The training procedure maximizes the cosine similarity between the RVQ layer outputs and the semantic teacher representations. HuBERT Layer 9 units represent semantic tokens, while EnCodec codes represent acoustic tokens.

## 3.2 DOWNSTREAM MODEL

In this step, we employ neural networks to solve supervised tasks of common interest. To achieve this, we first assign each discrete token to a corresponding embedding vector through a lookup table. Subsequently, we dynamically combine the embeddings from different codebooks using attention, enabling the model to adjust the importance of codebooks for the specific task of interest. The attention mechanism consists of a simple multi-layer perceptron (MLP) that takes the embeddings of the audio tokens as input. The MLP generates a score for each selected codebook, which is

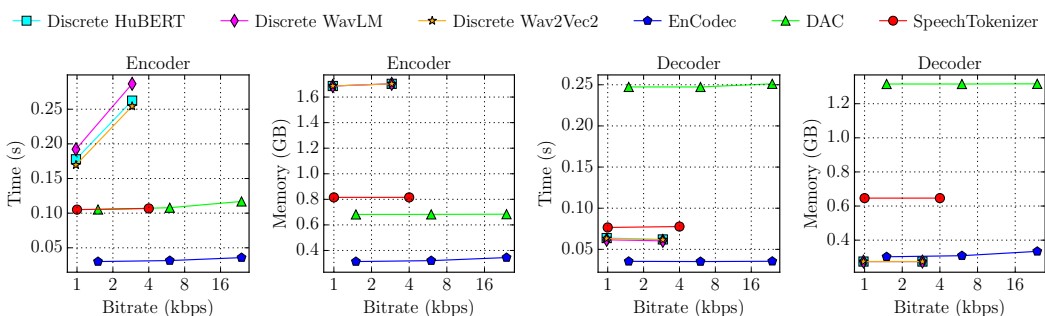

Figure 2: Time and memory required to process an utterance of 16 seconds for encoders and decoders of the considered audio tokenizers on an NVIDIA GeForce RTX 3070 GPU @ 8 GB.

normalized by a softmax function as shown in the following equations:

$$
z_{c,t} = f\big(\text{emb}(d_{c,t})\big), \ a_{c,t} = \frac{\exp(z_{c,t})}{\sum_{k=1}^{C} \exp(z_{k,t})}, \ h_t = \sum_c a_{c,t} z_{c,t}, \tag{2}
$$

where $d_{c,t}$ is the discrete token obtained from codebook $c$, at time $t$ and, $z_{c,t}$ represents the score assigned to codebook $c$ at time $t$ by the MLP function $f$. The function $\text{emb}(\cdot)$ refers to the lookup table that assigns embeddings to each discrete token. The variable $a_{c,t}$ denotes the attention assigned to the codebook $c$ at time $t$, and lastly $h_t$ is the representation that is fed to the downstream MLP model. We would like to note that the MLP learns different codebook combinations $h_t$ at each time step $t$ depending on the attention weights $a_{c,t}$, enabling the model to extract the necessary information when required. We also considered summing all embeddings, as has been done in previous literature Puvvada et al. (2024); Wu et al. (2024), but found that attention weights performed slightly better.

The combined representations $h_t$ are fed into neural models designed for different tasks. The downstream models are jointly trained with their attention and embedding layers in an end-to-end fashion. For discriminative tasks, the model outputs either a single prediction (e.g., for emotion recognition) or a sequence of predictions (e.g., for speech recognition). For generative tasks (e.g., speech enhancement), the neural network outputs the targeted tokens per codebook, with the output shape being $C \times L$, where $L$ is the sequence length. The predicted audio tokens are converted into an audio waveform via an audio decoder, as explained in the next section.

Our benchmark aims to assess audio tokens using relatively simple downstream architectures. The rationale behind this approach is to ensure that the evaluation focuses on the quality of the audio tokens themselves. If the downstream architectures were too sophisticated, they could potentially "compensate" for weaker audio tokens, which would impair a fair comparison. This approach is consistent with the SUPERB wen Yang et al. (2021) benchmark used for evaluating speech self-supervised models, and other benchmarks (e.g., USB Wang et al. (2022) and MP3S Zaiem et al. (2023)). Our intent is thus not to claim state-of-the-art results for specific downstream tasks but rather to provide a fair comparison of different tokenizers. By using a simple architecture for all tokenizers, we maintain the fairness of the comparison.

### 3.3 DISCRETE AUDIO DECODER

The decoder, used for generative tasks only, converts the predicted tokens into audio signals. The decoder is frozen during training. The choice of decoder depends on the encoder used in the first step. For compression and hybrid tokenizers, we use their built-in decoder. For semantic tokens, we use the scalable vocoder proposed in Mousavi et al. (2024), which is a modified HiFi-GAN Yang et al. (2023) pretrained with LibriSpeech-960h Korvas et al. (2014). The scalable vocoder accepts a variable number of multi-layer semantic tokens as input and can handle different bitrates using a layer dropout mechanism.

## 4 EXPERIMENTS

In the following sections, we describe the discriminative and generative tasks considered in our experiments. For a more reliable evaluation, we tested two downstream architectures for each task. For detailed information about the hyperparameters used in each experiment, refer to Appendix **??**.

### 4.1 DISCRIMINATIVE TASKS

For the downstream architectures and training procedures, we follow the best-performing approaches for classic continuous self-supervised representations proposed in Zaiem et al. (2023).

- **Automatic Speech Recognition (ASR)**: The goal of ASR is converting speech signals into written text. We address two ASR tasks. The first task involves English ASR using the popular LibriSpeech dataset Korvas et al. (2014). Training and validation are performed on the train-clean-100 and dev-clean subsets, respectively, while testing is conducted on the test-clean and test-other subsets. The downstream architecture for this task consists of two layers of Bidirectional Long Short-Term Memory (BiLSTM) followed by a linear layer for mapping audio to characters. The second architecture utilizes ContextNet Han et al. (2020) with unitary strides to maintain the frame rate of the encoder models. Additionally, we explore low-resource languages, specifically Welsh (Cymraeg) and Basque (Euskera) datasets extracted from CommonVoice 17.0 Ardila et al. (2020). Here, we evaluate the performance using both the BiLSTM architecture and a two-layer dense neural network mapping frame representations to character probabilities. We use the Word Error Rate (WER) as the error metric for all ASR tasks.

- **Speaker Identification/Verification (SID, SV)**: Speaker Identification involves classifying each utterance by its speaker identity as a multi-class classification, with the same predefined set of speakers for both training and testing. The evaluation metric is the accuracy. Automatic Speaker Verification (ASV), instead, involves training a binary classifier to determine whether the speakers in a pair of utterances are the same. The evaluation metric adopted in this case is the equal error rate (EER). We use the widely-used VoxCeleb1 Nagrani et al. (2017) train and test splits for both tasks. First, we test the X-vector Snyder et al. (2018) architecture with AM-Softmax Wang et al. (2018) loss for training the speaker embeddings. For verification, we use the cosine similarity between speaker representations. As a second architecture, we replace the X-vectors with an ECAPA-TDNN neural network Desplanques et al. (2020).

- **Emotion Recognition (ER)**: The task involves predicting one of the four classes: *happy*, *sad*, *angry*, and *neutral*. We use the popular IEMOCAP Busso et al. (2008) dataset, which contains about 10k samples from 10 speakers. As a first architecture, we directly input the representations into a linear classification layer after averaging them along the time axis. For the second downstream architecture, we use ECAPA-TDNN. The evaluation metric is the accuracy.

- **Intent Classification (IC)**: This task aims to determine the intention or purpose given utterance a speech recording. In particular, we here aim to classify each utterance into one of 18 scenarios, including *calendar*, *email*, and *alarm*. For this task, we utilize the SLURP dataset Bastianelli et al. (2020), which comprises around 72k audio recordings of single-turn user interactions with a home assistant. We employ ECAPA-TDNN and a two-layer BiLSTM (followed by a linear classifier) as downstream architectures. We evaluate the performance using the accuracy.

- **Keyword Spotting (KS)**: Keyword Spotting involves detecting predefined keywords by classifying utterances into a set of specified words. We use the Speech Commands dataset v1.0 Warden (2018) for this task, as done in SUPERB. The dataset includes ten classes of keywords, a class for silence, and an unknown class to account for false positives. We employ both the X-vector and ECAPA-TDNN architectures. The evaluation metric is the accuracy.

- **Event Sound Detection (ES)**: This task involves classifying audio clips into 50 different sound event categories, using the ESC-50 dataset Piczak (2015), which contains 2,000 labeled 5-second recordings across 50 sound classes (40 samples per class). We employed two downstream architectures: linear and ECAPA-TDNN, with accuracy as the evaluation metric. To compare semantic tokenizers, we used self-supervised models trained on audio, contrasting them with compression-based tokens from large sound and music datasets. For sound classification, we utilized the pre-trained CNN14 model Kong et al. (2020) (trained on VGGSound with SimCLR) and applied k-means clustering across four layers for medium bitrate, and two layers for low bitrate.

- **Music Genre Classification (MG)**: This task focuses on classifying music clips into 10 genre categories using the GTZAN dataset Tzanetakis & Cook (2002), which contains 100 audio files per genre, each 30 seconds long. Similar to event sound detection, we used two downstream architectures: linear and ECAPA-TDNN, with accuracy as the evaluation metric. For semantic tokenizers, we used the MERT Li et al. (2023) model, which has a similar architecture to wav2vec2. The same bitrate settings—low and medium—used for discrete self-supervised models in speech tasks were applied here.

## 4.2 Generative Tasks

- **Speech Enhancement (SE)**: Speech enhancement aims to improve audio quality by cleaning up noisy input recordings. For this task, we utilize the popular VoiceBank dataset Valentini-Botinhao et al. (2016). We employ two downstream architectures: a non-autoregressive Conformer encoder Gulati et al. (2020), and a convolutional recurrent deep neural network (CRDNN). The input tokens are extracted from the noisy signal, while target tokens from the clean one. Training is performed using the cross-entropy loss. The speech quality is assessed using the deep noise suppression mean opinion score (DNSMOS) Reddy et al. (2022). The intelligibility is evaluated through the differential word error rate (dWER) Wang et al. (2021), which measures the WER between the transcribed enhanced signal and the transcribed target signal. The transcriptions are obtained using the small version of Whisper Radford et al. (2022). Additionally, to measure speaker fidelity, we use the cosine similarity (SpkSim) between X-vectors extracted from the enhanced signal and the target signal using the base variant of WavLM Chen et al. (2022) fine-tuned for speaker verification.

- **Speech Separation (SS)**: Speech separation aims to isolate individual voices from an audio recording containing multiple speakers. For this task, we use the Libri2Mix dataset Cosentino et al. (2020), which contains mixtures of two overlapping speakers. We employ two downstream architectures: a non-autoregressive Conformer encoder Gulati et al. (2020), and a convolutional recurrent deep neural network (CRDNN). The input tokens are extracted from the mixture, while target tokens from each of the two sources. Training is performed using the cross-entropy loss with permutation invariant training Kolbæk et al. (2017). To measure performance, we employ the same metrics as speech enhancement.

- **Text-to-Speech (TTS)**: The task involves generating raw speech audio from a given text input. For downstream architectures, we consider both a small and a large autoregressive Transformer Vaswani et al. (2017). We train all models on the LJSpeech dataset Ito (2017). To assess the speech quality, we use a pretrained UTokyo-SaruLab System (UTMOS) Takaaki et al. (2022), which is specifically designed for TTS and trained to predict human Mean Opinion scores. To measure pronunciation accuracy, we use the dWER metric. This involves comparing the transcriptions provided by a speech recognizer for the synthesized speech sample with the transcriptions from the ground truth. As for enhancement and separation, we considered the small version of Whisper Radford et al. (2022).

## 5 Results

### 5.1 Comparison of Discrete Audio Models

Tables 2 and 3 show the performance of discriminative and generative tasks among the two downstream architectures explored. For each value in the table, we report the best performance obtained with the two downstream architectures. Detailed results for each architecture, along with the settings for each experiment using continuous SSL models, are provided in Appendix E.

We observe a significant variation in the tokenizer performance across different tasks. This result suggests that the optimal choice of tokenizer depends on the specific task at hand. However, some interesting patterns emerge. For instance, semantic tokens significantly outperform compression tokens for most discriminative tasks. This trend is due to the ability of semantic tokens to capture high-level information from the audio signal as also observed in existing findings in the literature Borsos et al. (2023). The only exceptions are speaker recognition tasks, where EnCodec achieves the best results. This suggests that compression tokens better encode speaker information. It is consistent with a previous study van Niekerk et al. (2022) that shows discrete tokens obtained from quantization of SSL layers remove speaker information.

Table 2: Benchmarking results for discriminative tasks.

| Models/Tasks | ASR-En | | ASR-multiling | | ER | IC | KS | SI | SV |
| | WER ↓ | | WER ↓ | | ACC ↑ | ACC ↑ | ACC ↑ | ACC ↑ | EER ↓ |
| | Clean | Other | Welsh | Basque | | | | | |
| *Low Bitrate* | | | | | | | | | |
| Discrete Hubert | **8.99** | **21.14** | **58.50** | **26.83** | 57.20 | 68.70 | 90.54 | 0.90 | 24.99 |
| Discrete WavLM | 11.72 | 27.56 | 60.37 | 28.63 | **59.80** | 73.40 | **97.94** | 0.70 | 26.02 |
| Discrete Wav2Vec2 | 12.14 | 28.65 | 66.30 | 32.25 | 57.80 | **74.10** | 96.16 | 0.40 | 33.53 |
| EnCodec | 52.37 | 77.04 | 92.01 | 58.20 | 44.70 | 31.50 | 86.00 | **58.30** | **17.40** |
| DAC | 63.96 | 83.61 | 94.86 | 66.29 | 49.20 | 22.10 | 81.00 | 45.10 | 20.62 |
| SpeechTokenizer | 19.77 | 43.12 | 76.67 | 47.92 | 49.10 | 57.90 | 95.09 | 47.40 | 20.41 |
| *Medium Bitrate* | | | | | | | | | |
| Discrete Hubert | **7.91** | **18.95** | 54.77 | 23.63 | **62.10** | 70.50 | 94.69 | 67.40 | 15.71 |
| Discrete WavLM | 8.52 | 20.35 | **54.22** | **22.06** | 57.60 | **78.00** | **98.09** | 80.80 | 8.00 |
| Discrete Wav2Vec2 | 8.76 | 21.32 | 60.39 | 26.64 | 59.10 | 75.10 | 96.64 | 65.47 | 17.64 |
| EnCodec | 46.80 | 74.24 | 91.23 | 47.95 | 51.30 | 31.40 | 88.70 | **91.90** | **7.81** |
| DAC | 59.54 | 81.48 | 97.43 | 56.16 | 45.80 | 18.90 | 76.60 | 83.80 | 11.78 |
| SpeechTokenizer | 18.32 | 41.21 | 75.17 | 38.94 | 52.10 | 57.80 | 94.86 | 91.40 | 7.88 |
| *High Bitrate* | | | | | | | | | |
| EnCodec | **45.18** | **72.56** | **93.40** | **87.65** | 46.40 | **19.60** | **83.60** | **92.81** | **7.18** |
| DAC | 99.53 | 99.38 | 99.40 | 99.68 | **46.00** | 15.70 | 75.20 | 85.61 | 10.89 |
| *Continuous Baseline* | | | | | | | | | |
| SSL | 3.370 | 7.04 | 41.77 | 14.32 | 63.10 | 86.10 | 99.00 | 99.70 | 2.10 |

Table 3: Benchmarking results for generative tasks. N.C. indicates "Not Converged".

| Models/Tasks | SE | | | SS | | | TTS | |
| | DNSMOS ↑ | dWER ↓ | SpkSim ↑ | DNSMOS ↑ | dWER ↓ | SpkSim ↑ | UTMOS ↑ | dWER ↓ |
| *Low Bitrate* | | | | | | | | |
| Discrete HuBERT | 3.33 | **15.47** | 0.824 | 3.52 | 80.86 | 0.840 | 3.24 | **2.55** |
| Discrete WavLM | 3.26 | 16.52 | 0.830 | 3.43 | **62.34** | 0.847 | **3.84** | 3.01 |
| Discrete Wav2Vec2 | **3.55** | 18.86 | 0.779 | **3.75** | 96.70 | 0.787 | 3.32 | 3.45 |
| EnCodec | 3.15 | 34.35 | 0.852 | 3.11 | 83.55 | **0.877** | 1.46 | 8.85 |
| DAC | 3.30 | 57.41 | 0.853 | 3.01 | 102.00 | 0.854 | 1.97 | 10.68 |
| SpeechTokenizer | 3.18 | 30.13 | **0.858** | 3.13 | 85.25 | 0.874 | 2.51 | 3.69 |
| *Medium Bitrate* | | | | | | | | |
| Discrete HuBERT | 3.48 | 12.62 | 0.875 | 3.70 | 66.29 | 0.891 | 3.80 | 3.40 |
| Discrete WavLM | 3.48 | **10.18** | 0.889 | 3.68 | **34.03** | 0.912 | **3.82** | **2.45** |
| Discrete Wav2Vec2 | **3.54** | 17.60 | 0.858 | **3.75** | 78.42 | 0.866 | 3.68 | 2.89 |
| EnCodec | 3.10 | 19.07 | 0.885 | 3.09 | 48.57 | 0.906 | 1.50 | 94.6 |
| DAC | 3.49 | 31.14 | **0.906** | 3.26 | 55.43 | **0.924** | 1.71 | 71.26 |
| SpeechTokenizer | 3.49 | 23.44 | 0.876 | 3.42 | 60.75 | 0.906 | 1.96 | 53.26 |
| *High Bitrate* | | | | | | | | |
| EnCodec | 2.87 | 68.22 | 0.814 | **2.95** | 97.73 | **0.839** | N.C | N.C |
| DAC | **2.95** | **46.07** | **0.860** | 2.53 | 208 | 0.784 | N.C | N.C |
| *Continuous Baseline* | | | | | | | | |
| SSL | 3.49 | 4.92 | 0.928 | 3.68 | 9.97 | 0.939 | 3.71 | 2.94 |

Semantic tokens show the best performance for generative tasks as well, achieving the highest MOS and dWER scores, indicating better overall quality and intelligibility in the generated outputs. However, for preserving speaker identity, compression tokens are more effective, as shown by superior speaker similarity (SpkSim) metrics. We think our findings for generative tasks are particularly interesting. While prior research efforts Wang et al. (2024); Mousavi et al. (2024) explored the use of semantic tokens for generation, they did not include a comparison with the performance of compression tokens. It is important to remark that the success observed with both semantic tokens relies heavily on the effectiveness of the decoder architecture used in our benchmark. Our scalable decoder minimizes distortions and artifacts in the generated speech, leading to better performance on various generative tasks.

In Table 4 (right), we present the ranking aggregation for the considered tokenizers (medium bitrate). Each model is individually ranked for every task, and we compute the average position across all ranks. This analysis shows that discrete WavLM generates the top-performing audio tokens.

Table 4: **(left)** Evaluating various discrete decoders on the speech re-synthesis task (medium bitrate). **(right)** Ranking aggregation for models (medium bitrate).

| Models/Metrics | DNSMOS ↑ | dWER ↓ | SpkSim ↑ |
|---|---|---|---|
| Discrete HuBERT | 3.68 | 6.60 | 0.92 |
| Discrete WavLM | 3.64 | 5.19 | 0.94 |
| Discrete Wav2Vec2 | 3.71 | 8.72 | 0.91 |
| EnCodec | 3.54 | **2.16** | 0.98 |
| DAC | **3.74** | 2.36 | **0.99** |
| SpeechTokenizer | 3.58 | 5.12 | 0.94 |
| Continuous SSL | 3.73 | 2.33 | 0.98 |

| Model | Disc. | Gen. | Comb. |
|---|---|---|---|
| Discrete HuBERT | 2.66 | 3.62 | 3.11 |
| Discrete WavLM | **2.00** | 2.75 | **1.94** |
| Discrete Wav2Vec2 | 3.33 | **2.68** | 3.41 |
| EnCodec | 4.11 | 3.93 | 4.23 |
| DAC | 5.55 | 4 .06 | 4.64 |
| SpeechTokenizer | 3.44 | 3.81 | 3.64 |

While the continuous version of WavLM ranks highest in the SUPERB benchmark, our findings demonstrate for the first time that this model maintains strong performance even after tokenization.

Our comparison between discrete tokens and the best continuous baseline reveals a significant performance gap favoring continuous representations. This suggests that tokenization loses valuable information, such as phonetics, speaker identity, and emotion. Addressing this information loss is a key challenge for future generations of audio tokens.

## 5.2 IMPACT OF BITRATE

We also study the impact of different bitrates on the performance of the tokenizers. Tables 2 and 3 show that a medium bitrate achieves the best results for both discriminative and generative tasks. Interestingly, higher bitrates, when available (e.g., for EnCodec and DAC), tend to degrade performance. While higher bitrates can potentially preserve more information, they also increase the output dimensionality of the model, making the task more challenging to solve. In some cases, we found the task so challenging with high bitrates that the model did not converge, as observed in the case of TTS. It is worth noting that semantic tokens have a lower bitrate than compression tokens, as shown in Table 1. For example, in the medium bitrate range, discrete WavLM has a bitrate of 2.9 kbps, while EnCodec has 6.0 kbps. This difference is due to the varying number of codebooks (6 vs. 8) and sampling rates (16 kHz vs. 24 kHz). Despite their lower bitrate, semantic tokens provide better performance.

Another aspect we investigate is the efficiency of the encoders and decoders, as this could impact some applications. Figure 2 shows the time and memory usage for each encoder-decoder pair across all bitrate ranges. For semantic tokens, the encoder is a large neural network and is computationally demanding. In contrast, the decoders are based on a compact HiFi-GAN model and are very efficient. For streaming tasks Wu et al. (2023) where time and memory are critical, EnCodec turned out to be a better candidate due to the efficiency of both its encoder and decoder models.

## 5.3 ANALYSIS OF DISCRETE AUDIO DECODER

Finally, we present a comparative evaluation of the decoders in Table 4 (left). The decoder evaluation is conducted on the LibriSpeech test-clean subset using a speech re-synthesis task, where we extract the tokens from each discrete audio encoder and reconstruct the speech using the associated decoders. Then, we evaluate the reconstructed speech based on speaker similarity, Mean Opinion Score (MOS), and differential Word Error Rate (dWER). The goal of this experiment is to establish if a given system is able to provide a high-fidelity reconstruction of the input audio after encoding it in the discrete space. This is especially important to establish for generative tasks.

The results show that the built-in decoders of compression tokens outperform other models in preserving speaker similarity, further confirming that current semantic tokens do not adequately preserve speaker information. Compression-based tokens also achieve better dWER scores. However, in terms of speech quality (assessed with DNSMOS), there are no significant differences between semantic and compression-based tokens. This trend indicates that while semantic tokens produce good-quality audio, they may be slightly more prone to semantic degradation (e.g., mispronunciations of words or phonemes). As expected, continuous baselines perform better than their discrete counterparts. Additional analysis on low and high settings can be found in Appendix D.

## 5.4 SOUND AND MUSIC TASKS

We extend our evaluations to include sound classification using the ESC50 dataset and music genre classification using the GTZAN dataset. Tables 5 and 6 present the detailed results of these experiments. The observed trend aligns with our findings in speech tasks: although a performance gap remains between the top discrete units and continuous SSL models, semantic tokenizers consistently outperform other tokenizers, with medium bitrate yielding the best results.

Table 5: Results for Event Sound Classification using ESC50.

| Models/Architectures | Linear ACC ↑ | ECAPA-TDNN ACC ↑ |
|---|---|---|
| *Low Bitrate* | | |
| Discrete CNN14 | **48.30** | **55.00** |
| EnCodec | 36.50 | 27.40 |
| DAC | 38.00 | 6.97 |
| SpeechTokenizer | 31.20 | 26.40 |
| *Medium Bitrate* | | |
| Discrete CNN14 | **65.50** | **53.10** |
| EnCodec | 38.00 | 9.86 |
| DAC | 31.50 | 12.50 |
| SpeechTokenizer | 37.30 | 26.70 |
| *High Bitrate* | | |
| EnCodec | **37.70** | **4.75** |
| DAC | 27.30 | 2.75 |
| *Continuous Baseline* | | |
| CNN14 Kong et al. (2020) | 72.10 | 67.50 |

Table 6: Results for Music Genre Classification using GTZAN.

| Models/Architectures | Linear ACC ↑ | ECAPA-TDNN ACC ↑ |
|---|---|---|
| *Low Bitrate* | | |
| Discrete MERT | **68.00** | **68.00** |
| EnCodec | 46.00 | 58.00 |
| DAC | 38.00 | 44.00 |
| SpeechTokenizer | 40.00 | 63.00 |
| *Medium Bitrate* | | |
| Discrete MERT | **71.00** | **76.00** |
| EnCodec | 49.00 | 56.00 |
| DAC | 37.00 | 48.00 |
| SpeechTokenizer | 35.00 | 65.00 |
| *High Bitrate* | | |
| EnCodec | **52.00** | **58.00** |
| DAC | 33.00 | 32.00 |
| *Continuous Baseline* | | |
| MERT Li et al. (2023) | 87.00 | 87.00 |

## 6 CONCLUSION

This paper introduces DASB, a comprehensive benchmark designed to evaluate the performance of discrete audio tokens across diverse tasks of broad interest. We employ various evaluation metrics, downstream architectures, and bitrates for more robust comparisons. Interestingly, our findings reveal that semantic tokens outperform, on average, compression tokens in both generative and discriminative tasks. In particular, discrete WavLM emerged as the top-performing model, making it a natural candidate for adoption in multi-modal text+audio LLMs. A significant performance gap, however, persists when compared to traditional self-supervised continuous representations. This highlights the need for further research, which we believe is essential for better incorporating audio tokens into large multimodal language models.

One limitation we encountered is the proprietary nature of some audio tokenizers, such as Soundstream Zeghidour et al. (2021), which are not publicly accessible. Additionally, the benchmark is currently limited to a few sound and music tasks, but we plan to further extend DASB to include more music and sound processing tasks, such as music generation. Our goal is to help the research community establish a shared benchmark and evaluation protocol for discrete audio representations. We will thus keep expanding DASB by continuously incorporating novel tokenizers and tasks.

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

## A  GENERAL INFORMATION

### A.1  COMPUTATIONAL RESOURCES

We designed our benchmark to be computationally accessible. Every task runs on GPUs with 32 GB or more of VRAM. Tasks like keyword spotting takes only 8 hours, while speech recognition (e.g, ASR-Basque) might require about 30 hours on a single NVIDIA V100 GPU.

### A.2  IMPACT

We believe DASB can have a positive impact on the research community. We do not foresee a direct negative societal impact or misuse of our benchmark. However, we acknowledge that DASB can potentially accelerate progress in multi-modal large language models, which, in turn, have a wide range of potential positive and negative uses that society is still working to assess.

### A.3  HOSTING AND MAINTENANCE PLAN

DASB platform is hosted and version-tracked via GitHub. DASB is a community-driven and open-source initiative. We plan to extend it by running additional experiments and including new audio tokenizers and tasks. We welcome external contributors.

### A.4  LICENSING

Our work is licensed under Apache 2.0 (https://www.apache.org/licenses/LICENSE-2.0).

Table 7: Licenses for the models used in our benchmark.

| Model | License |
|---|---|
| HuBERT-large | Apache 2.0 |
| WavLM-large | ATTRIBUTION-SHAREALIKE 2.0 |
| Wav2vec2-large | Apache 2.0 |
| EnCodec | MIT license |
| DAC | MIT license |
| SpeechTokenizer | Apache 2.0 |

### A.5  AUTHOR STATEMENT

We, the authors, will bear all responsibility in case of violation of rights.

## B   DISCRETE AUDIO MODELS DETAILS

Table 8: Features of the Considered Discrete Audio Encoders.

| Model | Dataset | Repo |
|---|---|---|
| Discrete Hubert Mousavi et al. (2024) | LibriSpeech960Korvas et al. (2014) | huggingface.co/speechbrain/SSL_Quantization |
| Discrete WavLM Mousavi et al. (2024) | LibriSpeech960Korvas et al. (2014) | huggingface.co/speechbrain/SSL_Quantization |
| Discrete Wav2Vec2 Mousavi et al. (2024) | LibriSpeech960Korvas et al. (2014) | huggingface.co/speechbrain/SSL_Quantization |
| EnCodec Défossez et al. (2023) | DNS Dubey et al. (2024), CommonVoice Ardila et al. (2020), AudioSet Gemmeke et al. (2017), FSD50K Fonseca et al. (2021), and Jamendo Bogdanov et al. (2019) | github.com/facebookresearch/encodec |
| DAC Kumar et al. (2023) | DAPSMysore (2014), DNS Dubey et al. (2024), CommonVoice Ardila et al. (2020), VCTK Yamagishi et al. (2019), MUSDB Rafii et al. (2017), and Jamendo Bogdanov et al. (2019) | github.com/descriptinc/descript-audio-codec |
| SpeechTokenizer Zhang et al. (2024) | LibriSpeech960Korvas et al. (2014) | github.com/ZhangXInFD/SpeechTokenizer |

## C   DATASET AND DOWNSTREAM MODELS

Table 9 provides a summary of the datasets and the two downstream architectures used for each task.

Table 9: Dataset and Downstream Models

| Dataset | Task | 1st Architecture | 2nd Architecture | Dataset Link |
|---|---|---|---|---|
| LibriSpeech Korvas et al. (2014) | Speech Recognition | BiLSTM | ContextNet | openslr.org/12 |
| CommonVoice 17.0 Ardila et al. (2020) | Speech Recognition | BiLSTM | Linear | commonvoice.mozilla.org/en/datasets |
| VoxCeleb1 Nagrani et al. (2017) | speaker verification/identification | ECAPA-TDNN | X-Vectors | robots.ox.ac.uk/ vgg/data/voxceleb/vox1.html |
| IEMOCAP Busso et al. (2008) | Emotion Recognition | ECAPA-TDNN | Time-Pooling + Linear | sail.usc.edu/iemocap/ |
| Speech Commands Warden (2018) | Keyword Spotting | X -Vectors | ECAPA-TDNN | tensorflow.org/datasets/catalog/speech_commands |
| SLURP Bastianelli et al. (2020) | Intent Classification | BiLSTM + Linear | Time-Pooling + Linear | zenodo.org/record/4274930 |
| VoiceBank Valentini-Botinhao et al. (2016) | Speech Enhancement | Conformer | CRDNN | datashare.ed.ac.uk/handle/10283/2791 |
| Libri2Mix Cosentino et al. (2020) | Speech Separation | Conformer | CRDNN | github.com/JorisCos/LibriMix |
| LJSpeech Ito (2017) | Text-to-Speech | Shallow Transformer | Deep Transformer | keithito.com/LJ-Speech-Dataset/ |
| ESC50 Piczak (2015) | Sound Classification | Linear | ECAPA-TDNN | https://github.com/karolpiczak/ESC-50 |
| GTZAN Tzanetakis & Cook (2002) | Music Genre Classification | Linear | ECAPA-TDNN | https://huggingface.co/datasets/marsyas/gtzan |

# D ADDITIONAL ANALYSIS OF DISCRETE AUDIO DECODERS

In this section, we expand on the results from Section 5.3 by including low and high bitrates in addition to the medium bitrate. Additionally, for speaker similarity, we measure the cosine similarity between X-vectors extracted from the reconstructed and target signals using two different models: WavLM (SpkSim WavLM) and ECAPA-TDNN (SpkSim ECAPA), both fine-tuned for speaker verification. When analyzing low and high bitrate settings, distinct trends emerge compared to the medium bitrate. At low bitrates, the models perform worse in terms of speaker similarity, MOS, and dWER, as expected. This is particularly pronounced for compression-based decoders, where the degradation is more significant in terms of dWER. In contrast, at high bitrate, there is an overall small improvement in all metrics. In particular, the DAC model consistently outperforms EnCodec across all evaluated metrics, even for high bitrate settings.

The analysis shows that bitrate significantly impacts the performance of discrete decoders. Higher bitrates better preserve speech characteristics and result in lower error rates, while lower bitrates degrade these aspects. This highlights the trade-off between bitrate and speech synthesis quality.

Table 10: Evaluation of various discrete decoders for the speech re-synthesis task at low, medium, and high bitrates.

| Models/Metrics | SpkSim ECAPA ↑ | SpkSim WavLM ↑ | DNSMOS ↑ | dWER ↓ |
|---|---|---|---|---|
| *Low Bitrate* | | | | |
| Discrete Hubert | 0.34 | 0.87 | 3.51 | 8.25 |
| Discrete WavLM | 0.34 | 0.88 | 3.45 | **7.01** |
| Discrete Wav2Vec2 | 0.28 | 0.82 | **3.73** | 10.58 |
| EnCodec | 0.52 | **0.92** | 3.20 | 11.71 |
| DAC | **0.54** | 0.91 | 3.42 | 14.02 |
| SpeechTokenizer | 0.22 | 0.72 | 3.21 | 45.7 |
| *Medium Bitrate* | | | | |
| Discrete Hubert | 0.47 | 0.92 | 3.68 | 6.60 |
| Discrete WavLM | 0.53 | 0.94 | 3.64 | 5.19 |
| Discrete Wav2Vec2 | 0.43 | 0.91 | 3.71 | 8.72 |
| EnCodec | **0.87** | 0.98 | 3.54 | **2.16** |
| DAC | **0.87** | **0.99** | **3.74** | 2.36 |
| SpeechTokenizer | 0.65 | 0.94 | 3.58 | 5.12 |
| *High Bitrate* | | | | |
| EnCodec | 0.94 | 0.99 | 3.69 | 1.47 |
| DAC | **0.98** | **100** | **3.79** | **0.73** |

# E  ADDITIONAL RESULTS

Tables 11 - 14 show the results obtained with 2 different downstream architectures. Note that table 9 indicates the first and second architectures explored for each task. For the continuous baseline, we follow the same architecture as the discrete experiments except for TTS. For the TTS continuous baseline, we use a modified Tacotron2 Shen et al. (2018) architecture enhanced with guided attention Tachibana et al. (2018) that predicts SSL representations instead of Mel spectrograms.

Varying the architecture of the downstream decoder leads to significant variations in task performance. For ASR tasks, BiLSTM performs better. For classification tasks, ECAPA-TDNN shows the best performance, except for keyword spotting where X-vector is slightly better. For speech enhancement and separation, Conformer shows the best performance. For TTS, a notable pattern is observed: semantic tokens yield the best results with shallow models, while acoustic and hybrid tokens perform better with deeper models but still underperform compared to semantic tokens. One reason might be that discrete SSL models retain higher-level features closer to phonetic transcriptions, requiring lower-capacity models to capture the relationship between raw text and such representations. Higher-capacity models can lead to slower training and potential overfitting. In contrast, shallow models appear to underfit acoustic tokens, resulting in high dWERs and speech-like sounds with only surface resemblance to the original sentence, rather than intelligible speech.

Table 11: Results for discriminative tasks with the first downstream architecture.

| Models/Tasks | ASR-En | | ASR-multiling | | ER | IC | KS | SI | SV |
|---|---|---|---|---|---|---|---|---|---|
| | WER ↓ | | WER ↓ | | ACC ↑ | ACC ↑ | ACC ↑ | ACC ↑ | EER ↓ |
| | Clean | Other | Welsh | Basque | | | | | |
| *Low Bitrate* | | | | | | | | | |
| Discrete Hubert | **8.99** | **21.14** | **58.50** | **26.83** | 57.20 | 68.70 | 90.54 | 0.90 | 24.99 |
| Discrete WavLM | 11.72 | 27.56 | 60.37 | 28.63 | **59.80** | 73.40 | **97.94** | 0.70 | 26.02 |
| Discrete Wav2Vec2 | 12.14 | 28.65 | 66.30 | 32.25 | 57.80 | **74.10** | 96.16 | 0.40 | 33.53 |
| EnCodec | 52.37 | 77.04 | 92.01 | 58.20 | 44.70 | 31.50 | 86.00 | **58.30** | **17.40** |
| DAC | 63.96 | 83.61 | 94.86 | 66.29 | 49.20 | 22.10 | 81.00 | 1.10 | 29.99 |
| SpeechTokenizer | 19.77 | 43.12 | 76.67 | 47.92 | 49.10 | 57.90 | 95.09 | 47.40 | 20.41 |
| *Medium Bitrate* | | | | | | | | | |
| Discrete Hubert | **7.91** | **18.95** | 54.77 | 23.63 | **62.10** | 70.50 | 94.69 | 67.40 | 15.71 |
| Discrete WavLM | 8.52 | 20.35 | **54.22** | **22.06** | 57.60 | **78.00** | **98.09** | 80.80 | 8.00 |
| Discrete Wav2Vec2 | 8.76 | 21.32 | 60.39 | 26.64 | 59.10 | 75.10 | 96.64 | 65.47 | 17.64 |
| EnCodec | 46.80 | 74.24 | 91.23 | 47.95 | 51.30 | 31.40 | 88.70 | **91.90** | **7.81** |
| DAC | 59.54 | 81.48 | 97.43 | 56.16 | 45.80 | 18.90 | 76.60 | 83.80 | 11.78 |
| SpeechTokenizer | 18.32 | 41.21 | 75.17 | 38.94 | 52.10 | 57.80 | 94.86 | 91.40 | 7.88 |
| *High Bitrate* | | | | | | | | | |
| EnCodec | **45.18** | **72.56** | **93.40** | **87.65** | 46.40 | **19.60** | **83.60** | **92.81** | **7.18** |
| DAC | 99.53 | 99.38 | 99.40 | 99.74 | **46.00** | 15.70 | 75.20 | 85.61 | 10.89 |
| *Continuous Baseline* | | | | | | | | | |
| SSL | 3.370 | 7.04 | 41.77 | 14.32 | 63.10 | 86.10 | 99.00 | 99.70 | 2.10 |

Table 12: Results for discriminative tasks with the second downstream architecture.

| Models/Tasks | ASR-En WER ↓ | | ASR-multiling WER ↓ | | ER ACC ↑ | IC ACC ↑ | KS ACC ↑ | SI ACC ↑ | SV EER ↓ |
|---|---|---|---|---|---|---|---|---|---|
| | Clean | Other | Welsh | Basque | | | | | |
| *Low Bitrate* | | | | | | | | | |
| Discrete Hubert | **11.99** | **23.45** | **97.30** | **99.20** | **61.40** | 57.80 | 90.54 | 15.60 | **18.34** |
| Discrete WavLM | 14.98 | 30.32 | 100.00 | 99.29 | 61.20 | 57.70 | **97.80** | 14.00 | 18.45 |
| Discrete Wav2Vec2 | 15.45 | 35.30 | 99.89 | 98.39 | 59.50 | **60.20** | 96.52 | 5.80 | 23.5830 |
| EnCodec | 90.84 | 94.97 | 99.88 | 100.00 | 40.20 | 18.40 | 88.50 | **34.70** | 23.08 |
| DAC | 125.00 | 119.00 | 100.00 | 100.00 | 47.20 | 17.70 | 83.90 | 21.50 | 27.92 |
| SpeechTokenizer | 26.50 | 47.91 | 100.00 | 99.11 | 50.40 | 49.10 | 94.55 | 24.50 | 25.92 |
| *Medium Bitrate* | | | | | | | | | |
| Discrete Hubert | **10.91** | **21.65** | 95.81 | 97.20 | **64.80** | 59.10 | 94.38 | 14.80 | 17.67 |
| Discrete WavLM | 11.12 | 22.63 | 96.51 | 97.38 | 60.60 | **63.80** | **97.85** | **25.60** | **15.74** |
| Discrete Wav2Vec2 | 12.48 | 25.40 | **94.34** | **95.76** | 61.90 | 63.00 | 96.92 | 11.70 | 19.07 |
| EnCodec | 124.00 | 119.00 | 99.82 | 99.98 | 41.50 | 18.20 | 89.10 | 30.10 | 21.73 |
| DAC | 124.00 | 122.00 | 99.96 | 100.00 | 46.80 | 16.40 | 80.30 | 15.90 | 29.75 |
| SpeechTokenizer | 118.00 | 117.00 | 97.08 | 96.49 | 56.60 | 47.60 | 89.10 | 32.80 | 20.15 |
| *High Bitrate* | | | | | | | | | |
| EnCodec | 124.00 | **122.00** | 100.00 | **99.93** | 43.40 | **17.10** | **80.40** | 23.40 | **25.52** |
| DAC | **122.00** | **122.00** | **99.74** | 100.00 | **47.70** | 15.40 | 76.00 | 16.40 | 29.94 |
| *Continuous Baseline* | | | | | | | | | |
| SSL | 10.05 | 13.80 | 68.72 | 48.60 | 68.60 | 75.20 | 98.70 | 88.40 | 4.31 |

Table 13: Results for generative tasks with the first downstream architecture. N.C. indicates "Not Converged".

| Models/Tasks | SE | | | SS | | | TTS | |
|---|---|---|---|---|---|---|---|---|
| | DNSMOS ↑ | dWER ↓ | SpkSim ↑ | DNSMOS ↑ | dWER ↓ | SpkSim ↑ | UTMOS ↑ | dWER ↓ |
| *Low Bitrate* | | | | | | | | |
| Discrete Hubert | 3.33 | **15.47** | 0.824 | 3.52 | 80.86 | 0.840 | 3.25 | **2.93** |
| Discrete WavLM | 3.26 | 16.52 | 0.830 | 3.43 | **62.34** | 0.847 | 2.74 | 12.69 |
| Discrete Wav2Vec2 | **3.55** | 18.86 | 0.779 | **3.75** | 96.70 | 0.787 | **3.32** | 3.45 |
| EnCodec | 3.15 | 34.35 | 0.852 | 3.11 | 83.55 | **0.877** | 1.46 | 8.84 |
| DAC | 3.30 | 57.41 | 0.853 | 3.01 | 102.00 | 0.854 | 1.97 | 10.66 |
| SpeechTokenizer | 3.18 | 30.13 | **0.858** | 3.13 | 85.25 | 0.874 | 2.55 | 16.42 |
| *Medium Bitrate* | | | | | | | | |
| Discrete HuBERT | 3.48 | 12.62 | 0.875 | 3.70 | 66.29 | 0.891 | **3.78** | 16.09 |
| Discrete WavLM | 3.48 | **10.18** | 0.889 | 3.68 | **34.03** | 0.912 | 3.77 | 14.76 |
| Discrete Wav2Vec2 | **3.54** | 17.60 | 0.858 | **3.75** | 78.42 | 0.866 | 3.68 | **5.98** |
| EnCodec | 3.10 | 19.07 | 0.885 | 3.09 | 48.57 | 0.906 | 1.43 | 92.4 |
| DAC | 3.49 | 31.14 | **0.906** | 3.26 | 55.43 | **0.924** | 1.71 | 71.25 |
| SpeechTokenizer | 3.49 | 23.44 | 0.876 | 3.42 | 60.75 | 0.906 | 1.97 | 53.26 |
| *High Bitrate* | | | | | | | | |
| EnCodec | 2.87 | 68.22 | 0.814 | **2.95** | 97.73 | **0.839** | N.C | N.C |
| DAC | **2.95** | **46.07** | **0.860** | 2.53 | 208 | 0.784 | N.C | N.C |
| *Continuous Baseline* | | | | | | | | |
| SSL | 3.49 | 4.92 | 0.928 | 3.68 | 9.97 | 0.939 | 2.86 | 4.687 |

Table 14: Results for generative tasks with the second downstream architecture. N.C. indicates "Not Converged".

| Models/Tasks | SE | | | SS | | | TTS | |
|---|---|---|---|---|---|---|---|---|
| | DNSMOS ↑ | dWER ↓ | SpkSim ↑ | DNSMOS ↑ | dWER ↓ | SpkSim ↑ | UTMOS ↑ | dWER ↓ |
| *Low Bitrate* | | | | | | | | |
| Discrete HuBERT | 3.31 | **13.98** | 0.821 | 3.52 | 97.58 | 0.817 | 3.24 | **2.55** |
| Discrete WavLM | 3.27 | 16.50 | 0.825 | 3.44 | **60.12** | 0.834 | **3.84** | 3.01 |
| Discrete Wav2Vec2 | **3.55** | 17.17 | 0.777 | **3.74** | 95.20 | 0.785 | 3.32 | 3.45 |
| EnCodec | 3.16 | 45.07 | 0.851 | 2.32 | 99.17 | 0.495 | 1.40 | 53.50 |
| DAC | 3.31 | 59.28 | **0.863** | 2.88 | 138 | 0.804 | 1.81 | 19.77 |
| SpeechTokenizer | 3.32 | 27.54 | 0.860 | 3.14 | 90.68 | **0.846** | 2.51 | 3.69 |
| *Medium Bitrate* | | | | | | | | |
| Discrete HuBERT | **3.48** | 13.71 | 0.857 | 3.72 | 91.63 | 0.843 | 3.80 | 3.40 |
| Discrete WavLM | 3.47 | **9.63** | **0.878** | 3.68 | **37.53** | **0.902** | 3.82 | **2.45** |
| Discrete Wav2Vec2 | 3.53 | 16.58 | 0.853 | **3.74** | 83.86 | 0.831 | 3.68 | 2.89 |
| EnCodec | 3.04 | 84.78 | 0.796 | 3.06 | 70.25 | 0.882 | 1.50 | 94.62 |
| DAC | 3.35 | 71.07 | 0.831 | 3.12 | 95.11 | 0.872 | 1.47 | 77.00 |
| SpeechTokenizer | 2.12 | 99.62 | 0.549 | 3.10 | 89.78 | 0.862 | 1.85 | 64.26 |
| *High Bitrate* | | | | | | | | |
| EnCodec | **2.93** | **95.06** | **0.780** | **2.96** | **157** | 0.768 | N.C | N.C |
| DAC | 2.22 | 99.62 | 0.571 | 2.31 | 234 | **0.787** | N.C | N.C |
| *Continuous Baseline* | | | | | | | | |
| SSL | 3.42 | 6.05 | 0.861 | 3.43 | 24.92 | 0.873 | 2.86 | 4.687 |

