# OpenReview forum: "DASB-Discrete Audio and Speech Benchmark"
_ICLR.cc/2025/Conference — Submitted to ICLR 2025_

### Official Review · Reviewer_8g5d · 2024-11-02

**Soundness:** 2
**Presentation:** 3
**Contribution:** 1
**Rating:** 3
**Confidence:** 5

**Summary:**

This paper introduces DASB, a benchmark for testing how well different architectures that generate "discrete" audio tokens perform in speech- and audio-related tasks. The benchmark include both discriminative tasks (e.g., ASR or classification) and generative ones (e.g., speech enhancement and TTS).

The paper clarify that the goal is to test different types of tokens, semantic-based, compression-based and hybrid. It tests several models on the benchmark in those three categories. For compression-based tokens it investigates how the bitrates affect token quality. In the results, however, there is no clear evidence of better models given by higher bitrates.

The main goal of the paper can be summarized as the definition of a standard ways to evaluate tokens generated by different SSL models that have been pretrained for different tasks and using different approaches (i.e., codec, wav2vec2-style mainly).

**Strengths:**

1. The paper addresses an important issue in emerging areas of Audio Representation Learning: evaluating the quality of representations, in this specific case given by discrete tokens.
2. The benchmark actually covers a broad set of speech-related tasks (e.g., ASR, classification with keyword spotting and intent classification, speaker identification and so on). It also include TTS and Speech Enhancement tasks as generative tasks.
3. The paper propose a straightforward explanation of each task and the architecture setup personalized by the paper for the task at hand. The paper clearly mention the dataset used for the evaluation.

**Weaknesses:**

1. The main goal of a benchmark framework should be to fairly assess the quality of the tested models. However, the current version of the evaluation design, while broad in terms of tasks, include multiple trainable components for each task making very hard to isolate the evaluation of the token quality. This issue is even more puzzling because the results are reported as a single run of a model instead of averaging multiple runs and reporting standard deviations. The results may be influenced by several factors that may hinder the real comparison.
2. The benchmark design involved selecting specific metrics for each task. While the selection for most of them is standard (e.g., accuracy for classification), in other cases it may need discussion (e.g., why not PESQ instead of DNSMOS for speech enhancement?).
3. There are several experimental details that are missing. For example, the extraction of discrete tokens from WavLM and HuBERT involves a K-Means step, what is the value of K used in the paper? It may significantly influence all the experiments together with all set of other parameters (e.g., batch size, learning rate...). Are those hyperparameters consistent for all models? In addition, it is not clear what is the hardware used to train the downstream models, Is it consistent for all models? The paper make reference to the RTX 3070 GPU (8GB of VRAM) for computing time and memory requirements but then states in Appendix A1 "Every task runs on GPUs with 32 GB or more of VRAM".
4. The benchmark fails to reference existing benchmark that evaluate audio representations. It makes reference to SUPERB, however, the benchmark position itself as a general one (Discrete **Audio** and Speech Benchmark). There are several examples of general (including speech) audio representation frameworks [1][2][3]. The paper fails to compare with them leaving an unclear picture of DASB's specific contributions. In most of the downstream tasks, the model under testing is obtaining tokens -- that are then mapped back to "vectors" using codebooks, is thus only a matter of how those models are pre-trained (or discretized in the case of WavLM and HuBERT) that changes? Why evaluating them in an existing benchmark is not sufficient? DASB includes ASR and generative tasks (e.g., speech enhancement and TTS), is this the only difference in addition to the evaluated models?
5. Concerning audio events, the paper states "For sound classification, we utilized the pre-trained CNN14 model...", this is quite a difference with respect to other transformer-based models used all along the paper. Transformer-based SSL models exist pre-trained on general audio datasets (e.g., AudioSet)[2][4] and they are available open-source.

[1] Turian, Joseph, et al. "Hear: Holistic evaluation of audio representations." NeurIPS 2021.
[2] M. La Quatra et al., "Benchmarking Representations for Speech, Music, and Acoustic Events", ICASSPW 2024.
[3] Wang, Luyu, et al. "Towards learning universal audio representations." ICASSP 2022.
[4] Gong, Y. et al "AST: Audio Spectrogram Transformer.", Proc. Interspeech 2021.

**Questions:**

1. How do you ensure that the benchmark isolates token quality when multiple trainable components are included?
2. Why were layers 1, 3, 7, 12, 18, and 23 selected for WavLM and HuBERT token extraction? This choice may appear arbitrary without reference to literature or specific validation.
3. The results lack practical takeaway. How do you plan the community to benefit from the insights obtained using DASB? There seems to be contrasting results (e.g., DAC vs Encodec) compared to recent papers [1] (e.g., on ESC-50); what could be the cause of these differences?
4. How DASB compares with existing benchmarks in the literature?
5. Do you notice any differences across multiple runs of the same model on specific DASB tasks? Can those differences influence the evaluation?
6. Are the experimental settings (in terms of hyperparameters) the same for all evaluated models?

[1] Ji, Shengpeng, et al. "Wavtokenizer: an efficient acoustic discrete codec tokenizer for audio language modeling." arXiv preprint arXiv:2408.16532 (2024).


Minor remarks:
- BERT is cited as an autoregressive model, how is it classified in this category?
- Mentioning Speech Enhancement with discrete tokens, the authors cited Deep Learning, the book from Goodfellow et al. (2016), the citation seems not pertinent.

---

> ### Author Response · Authors · 2024-11-25
>
> ## Weaknesses
>
> We thank the reviewer for their time and valuable feedback. We appreciate the constructive suggestions and are glad to receive insightful comments on our work. Following your feedback, we will update our paper accordingly.
>
> >***The main goal of a benchmark framework should be to fairly assess the quality of the tested models. However, the current version of the evaluation design, while broad in terms of tasks, include multiple trainable components for each task making very hard to isolate the evaluation of the token quality. This issue is even more puzzling because the results are reported as a single run of a model instead of averaging multiple runs and reporting standard deviations. The results may be influenced by several factors that may hinder the real comparison.***
>
> We agree that incorporating systematic hyperparameter tuning and averaging results over multiple runs would enhance the reliability of our findings. Although such adjustments could improve the metrics reported, we believe they might not significantly alter the overall patterns and rankings observed but could mitigate the large gaps. Similar to a good practice suggest for assessing self-supervised learning models proposed in [1], we are considering two different downstream architectures, which makes our results less noisy than what is often done.  We have already conducted some limited and manual hyperparameter tuning for each task. However, due to resource constraints and the extensive range of tasks and settings our benchmark covers, our ability to perform a more comprehensive analysis has been constrained. We are actively working to overcome these challenges. To strengthen the validity of our benchmark results, we are prioritizing improvements such as systematic hyperparameter tuning using the Bayesian Optimization and Hyperband (BOHB) algorithm for learning rates, which, according to our initial experiments, is one of the main factors especially for compression-based tokens.
> - [1]: Zaiem, Salah, et al. "Speech self-supervised representation benchmarking: Are we doing it right?." arXiv preprint arXiv:2306.00452 (2023).
>
>
> >***The benchmark design involved selecting specific metrics for each task. While the selection for most of them is standard (e.g., accuracy for classification), in other cases it may need discussion (e.g., why not PESQ instead of DNSMOS for speech enhancement?).***
>
> We chose DNSMOS as a metric for speech enhancement because it has been shown to be more reliable than other commonly used objective metrics like PESQ, SDR, and POLQA, as indicated in [1,3,4]. Notably, DNSMOS does not require reference clean speech, making it suitable for evaluating real-world recordings. Additionally, similarly to SI-SNR, intrusive metrics like PESQ and STOI may fail to accurately assess signal quality in the context of generative models. Generative models focus on modeling the distribution of real-world data, meaning their outputs can sound realistic even if they deviate significantly from the clean reference.
>
> Moreover, recent papers on generative speech enhancement [2,3] frequently utilize DNSMOS, reinforcing its relevance and applicability in current research.
> - [1] https://www.isca-archive.org/interspeech_2021/reddy21_interspeech.pdf
> - [2]Wang, Ziqian, et al. "SELM: Speech enhancement using discrete tokens and language models." ICASSP 2024-2024 IEEE International Conference on Acoustics, Speech and Signal Processing (ICASSP). IEEE, 2024.
> - [3] Xue, Huaying, Xiulian Peng, and Yan Lu. "Low-latency speech enhancement via speech token generation." ICASSP 2024-2024 IEEE International Conference on Acoustics, Speech and Signal Processing (ICASSP). IEEE, 2024.

---

> > ### Author Response · Authors · 2024-11-25
> >
> > ## Weaknesses
> > >***There are several experimental details that are missing. For example, the extraction of discrete tokens from WavLM and HuBERT involves a K-Means step, what is the value of K used in the paper? It may significantly influence all the experiments together with all set of other parameters (e.g., batch size, learning rate...). Are those hyperparameters consistent for all models? In addition, it is not clear what is the hardware used to train the downstream models, Is it consistent for all models? The paper make reference to the RTX 3070 GPU (8GB of VRAM) for computing time and memory requirements but then states in Appendix A1 "Every task runs on GPUs with 32 GB or more of VRAM".***
> >
> > For more detailed settings and hyperparameters, please refer to the attached code. This includes detailed documentation of all experimental parameters to ensure full transparency and reproducibility of our results. We will also release the code along with the camera-ready version of our paper.  However, we would like to use this opportunity to elaborate on the questions you asked.
> > - For all K-means clustering, we use 1000 clusters,  based on the observation from this paper [1].
> > - The hyperparameters remain consistent across all tokenizers.
> > - Regarding hardware, all models are trained under the same settings. For time and memory analysis, we utilize GPUs with 8GB of VRAM, which is adequate for processing an utterance of 16 seconds. For downstream tasks, we use GPUs with 32GB of VRAM. The choice of 8GB for time and memory analysis is based on its sufficiency for shorter tasks. Nevertheless, all time and memory evaluations are consistently performed using the same hardware specifications across all tokenizers.
> > - [1]:  Mousavi, Pooneh, et al. "How Should We Extract Discrete Audio Tokens from Self-Supervised Models?." arXiv preprint arXiv:2406.10735 (2024).
> >
> > >***Concerning audio events, the paper states "For sound classification, we utilized the pre-trained CNN14 model...", this is quite a difference with respect to other transformer-based models used all along the paper. Transformer-based SSL models exist pre-trained on general audio datasets (e.g., AudioSet)[2][4] and they are available open-source.
> >  [4] Gong, Y. et al "AST: Audio Spectrogram Transformer.", Proc. Interspeech 2021.***
> >
> > Thank you for your suggestion. We are in the process of incorporating BEATs[1] as a sound SSL model, as it also utilizes masked-based transformer models, aligning more consistently with the SSL choices for other domains featured in our paper. Additionally, we plan to expand the scope of DASB to include more tasks related to the music and sound domains, such as music generation, Singing Voice Synthesis, and Audio Question Answering.
> > - [1] Chen, Sanyuan, et al. "Beats: Audio pre-training with acoustic tokenizers." arXiv preprint arXiv:2212.09058 (2022).

---

> > > ### Author Response · Authors · 2024-11-25
> > >
> > > ## Weaknesses
> > > >***The benchmark fails to reference existing benchmarks that evaluate audio representations. It makes reference to SUPERB, however, the benchmark positions itself as a general one (Discrete Audio and Speech Benchmark). There are several examples of general (including speech) audio representation frameworks [1][2][3]. The paper fails to compare with them leaving an unclear picture of DASB's specific contributions. In most of the downstream tasks, the model under testing is obtaining tokens -- that are then mapped back to "vectors" using codebooks, is thus only a matter of how those models are pre-trained (or discretized in the case of WavLM and HuBERT) that changes? Why evaluating them in an existing benchmark is not sufficient? DASB includes ASR and generative tasks (e.g., speech enhancement and TTS), is this the only difference in addition to the evaluated models?
> > > [1] Turian, Joseph, et al. "Hear: Holistic evaluation of audio representations." NeurIPS 2021. [2] M. La Quatra et al., "Benchmarking Representations for Speech, Music, and Acoustic Events", ICASSPW 2024. [3] Wang, Luyu, et al. "Towards learning universal audio representations." ICASSP 2022.***
> > >
> > > As noted in the related works section, both Codec-SUPERB and ESPnet-Codec focus on evaluating the quality of resynthesized sound, utilizing compression and hybrid tokenizers. They have conducted a comprehensive study using pretrained models for speech, speaker, and emotion recognition to determine how much information the ***resynthesized audio*** retains. However, these studies have not directly addressed the use of tokenized inputs for training ***downstream tasks*** or analyzed the role of ***semantic tokens***. While  ESPnet-Codec addresses some downstream tasks, it does not offer as comprehensive a coverage as our DASB benchmark, and it employs more complex architectures. In both Codec-SUPERB and ESPnet-Codec, the good results may disproportionately reflect the capabilities of the decoders rather than the tokens themselves.
> > > Our DASB benchmark is designed specifically to assess the efficacy of audio tokens using relatively simple downstream architectures. This approach ensures the focus remains on the tokens' quality. By avoiding overly complex architectures, we prevent them from compensating for less effective tokens, thereby maintaining an equitable basis for comparison. This strategy aligns with established benchmarks like SUPERB[3], USB[1], and MP3S[2], which are all designed to evaluate the effectiveness of speech self-supervised models. Our intent is thus not to claim state-of-the-art results for specific downstream tasks but rather to explore how each tokenizer varies in its capacity to preserve specific information and identify its limitations
> > > - [1]: Wang, Yidong, et al. "Usb: A unified semi-supervised learning benchmark for classification." Advances in Neural Information Processing Systems 35 (2022): 3938-3961.
> > > - [2]: Zaiem, Salah, et al. "Speech self-supervised representation benchmarking: Are we doing it right?." arXiv preprint arXiv:2306.00452 (2023).
> > > - [3]: Yang, Shu-wen, et al. "Superb: Speech processing universal performance benchmark." arXiv preprint arXiv:2105.01051 (2021).

---

> > > > ### Author Response · Authors · 2024-11-25
> > > >
> > > > ## Questions:
> > > > >***How do you ensure that the benchmark isolates token quality when multiple trainable components are included?***
> > > >
> > > > Please refer to our answer in weakness section.
> > > >
> > > > >***Why were layers 1, 3, 7, 12, 18, and 23 selected for WavLM and HuBERT token extraction? This choice may appear arbitrary without reference to literature or specific validation.***
> > > > >
> > > > It is based on observation from prior research (WavLM ,Superb, How Should We Extract Discrete Audio Tokens from Self-Supervised Models?[1]  papers) which studied the contribution patterns of different layers across various tasks.
> > > > - [1] Mousavi, Pooneh, et al. "How Should We Extract Discrete Audio Tokens from Self-Supervised Models?." arXiv preprint arXiv:2406.10735 (2024).
> > > >
> > > > ***The results lack practical takeaway. How do you plan the community to benefit from the insights obtained using DASB? There seems to be contrasting results (e.g., DAC vs Encodec) compared to recent papers [1] (e.g., on ESC-50); what could be the cause of these differences? [1] Ji, Shengpeng, et al. "Wavtokenizer: an efficient acoustic discrete codec tokenizer for audio language modeling." arXiv preprint arXiv:2408.16532 (2024).***
> > > >
> > > > Please refer to our answer in the weakness section.
> > > >
> > > > >**How DASB compares with existing benchmarks in the literature?***
> > > >
> > > > Please refer to our answer in the weakness section.
> > > >
> > > > >***Do you notice any differences across multiple runs of the same model on specific DASB tasks? Can those differences influence the evaluation? And 6. Are the experimental settings (in terms of hyperparameters) the same for all evaluated models?***
> > > >
> > > > All settings are the same across all tokenizers for each task/downstream head. Indeed, incorporating systematic hyperparameter tuning and averaging results over multiple runs would enhance the reliability of our findings. Although such adjustments could improve the metrics reported, we believe they might not significantly alter the overall patterns and rankings observed.  Similar to a good practice suggest for assessing self-supervised learning models proposed in [1], we are considering two different downstream architectures, which makes our results less noisy than what is often done. We have already conducted some limited and manual hyperparameter tuning for each task. However, due to resource constraints and the extensive range of tasks and settings our benchmark covers, our ability to perform a more comprehensive analysis has been constrained.
> > > >
> > > > - [1]: Zaiem, Salah, et al. "Speech self-supervised representation benchmarking: Are we doing it right?." arXiv preprint arXiv:2306.00452 (2023).

---

> ### Comment · Reviewer_8g5d · 2024-11-26
>
> **hyperparameter tuning and results stability:**
>
> I must firmly disagree with your (first) response. It does not address the core issue I raised. My concern was not about hyperparameter tuning - indeed, hyp tuning could obfuscate the evaluation of intrinsic quality of the tokens. Rather, I specifically questioned the lack of multiple runs with fixed settings to assess the inherent variability of your evaluation framework.
> The citation provided (Zaiem et al. [1]) actually contradicts your argument. This paper actually states that different architectures lead to different rankings and shows poor correlations between results across architectures. This directly supports my first idea: your current evaluation framework, with its multiple trainable components, makes it impossible to isolate and fairly assess token quality. To improve the scientific validity of your benchmark, it requires to have multiple runs with identical settings to quantify result stability and teported std across runs.
>
> **K-Means clustering**
>
> Citation [1] to justify using 1000 clusters may not seem that correct. Its results (Table 1 for ASR EN, ASR FR, SID and ER) actually show that 2000 clusters perform better. This, to me, seems to contradict your choice and suggest arbitrary settings for these tasks. I would not ask to test all possible value, but did you evaluate this impact? This should be crucial for a benchmark.
> Also as a side note, deferring to "attached code" is not acceptable for relevant details that must be in the main body of the paper to be self-contained (while I strongly recognize open-source implementation as a crucial and relevant strong positive point of course).
>
> **Audio benchmarking, existing SSL models**
>
> The response fails to address the core issue. There already are transformer-based models (HuBERT, Wav2Vec2) that can be used for audio events, yet the shift to CNN14 seems arbitrary. This seems a critical inconsistency, especially when suitable transformer alternatives exist. As far as I understand, the addition of BEATs does not justify missing models in your current evaluation framework.
>
> **Existing benchmarks**
>
> The response completely misses my point and instead discusses Codec-SUPERB which I never mentioned. I specifically cited work on universal audio representations - all focused on evaluating audio representations generally, not codec quality.
> These benchmarks I suppose to be directly relevant to DASB since most of the proposed evaluation essentially tests representation quality (tokens are mapped back to vectors via codebooks, if I'm not wrong). Still the questions remains:
>
> - What unique insights does DASB provide beyond existing frameworks?
> - Why couldn't these tokens be evaluated using existing benchmarks?
> - How the DASB methodological choices compare to evaluation approaches in representation learnig (e.g. linear probing)?

---

### Official Review · Reviewer_XCwv · 2024-11-03

**Soundness:** 3
**Presentation:** 3
**Contribution:** 3
**Rating:** 6
**Confidence:** 3

**Summary:**

This paper presents DASB, a benchmark for evaluating discrete speech and audio units. DASB consists of several discriminative and generative tasks, including speech recognition, emotion recognition, speech enhancement, text-to-speech synthesis, etc. It provides a unified pipeline for leveraging different discrete units to perform these tasks. Preliminary evaluation results indicate that units derived from self-supervised learning models generally outperform codec units and that continuous representations significantly outperform discrete units.

**Strengths:**

This paper presents a benchmark for evaluating the direct use of discrete speech and audio units. The authors have constructed a unified pipeline along with open-source implementations, making it convenient for researchers to conduct various experiments using the benchmark. I anticipate that this benchmark will enable fair comparisons in future research on discrete speech and audio units. The evaluation results appear mostly reasonable. The presentation is clear and easy to read.

**Weaknesses:**

Although the authors present DASB as an audio and speech benchmark that includes both speech and audio tasks for evaluation, the number of audio tasks is rather limited, and there are fewer relevant descriptions regarding the development of audio and music units. Additionally, the models used in the experiments are limited. For example, I believe there have been several works on hybrid tokenizers, but in the experiment section, only SpeechTokenizer is included.

**Questions:**

1. In the abstract, the authors use the terms "semantic" and "compression" tokens, whose definitions do not appear until Section 3.1. Therefore, I was a bit confused when reading the abstract (though I could still guess to some extent). It would be helpful if the authors could elaborate more on this.

2. I believe there have been several studies on hybrid tokens this year, but only SpeechTokenizer is included. This type of token is a very active area of research. Is there a chance of providing more results in this category?

3. At the beginning of Section 4, there is an error in the Appendix index. Please fix it.

4. Which continuous SSL models are used exactly in Tables 2, 3, and 4?

5. In ASR, the authors include two low-resource languages. I wonder if there are any specific reasons for choosing them. The evaluation results on these two languages are very poor across the models, and I think that instead of directly comparing them at this level, using phoneme recognition could lower the difficulty of the task and provide more comparable results.

6. For speech generation tasks, the authors use automatic MOS prediction models to evaluate the quality of generated speech. I understand that this is mainly because MOS involves human effort, and it is infeasible to use only the original MOS in a benchmark. However, as far as I know, the correlation between automatic MOS models and human ratings is still limited. Could you provide the correlation of these results with human ratings?

7. In Tables 5 and 6, the discrete SSL parts are all replaced with CNN14 and MERT. Is there a chance to add results on the discrete HuBERT and WavLM? This could help test the generalizability from speech to audio and music.

---

> ### Author Response · Authors · 2024-11-25
>
> ## Weaknesses
>
> We thank the reviewer for their time and valuable feedback. We appreciate the constructive suggestions and are glad to receive insightful comments on our work. Following your feedback, we will update our paper accordingly.
>
> >***Although the authors present DASB as an audio and speech benchmark that includes both speech and audio tasks for evaluation, the number of audio tasks is rather limited, and there are fewer relevant descriptions regarding the development of audio and music units. Additionally, the models used in the experiments are limited. For example, I believe there have been several works on hybrid tokenizers, but in the experiment section, only SpeechTokenizer is included***
>
> We are actively working on expanding the benchmark by integrating more publicly available audio tokenizers and adding a broader array of tasks. Furthermore, we intend to broaden DASB's scope to include additional tasks related to music and sound domains, such as music generation, Singing Voice Synthesis, and Audio Question Answering. We would like to note that at the time of submitting this benchmark, many new tokenizers had not yet been released or officially accepted at a conference. However, we are actively enhancing our benchmark by integrating more publicly available audio tokenizers, including Mimi[1], SQ-Codec[2] (for scalar quantization), and WavTokenizer[3] (for single-layer quantization). This ensures that we include a diverse range of tokenizer categories to provide a comprehensive evaluation.
>
> - [1]: Défossez, Alexandre, et al. "Moshi: a speech-text foundation model for real-time dialogue." arXiv preprint arXiv:2410.00037 (2024).
> - [2]: Ji, Shengpeng, et al. "Wavtokenizer: an efficient acoustic discrete codec tokenizer for audio language modeling." arXiv preprint arXiv:2408.16532 (2024).
> - [3] Yang, Dongchao, et al. "Simplespeech 2: Towards simple and efficient text-to-speech with flow-based scalar latent transformer diffusion models." arXiv preprint arXiv:2408.13893 (2024).

---

> > ### Author Response · Authors · 2024-11-25
> >
> > ## Questions:
> >
> > >***In the abstract, the authors use the terms "semantic" and "compression" tokens, whose definitions do not appear until Section 3.1. Therefore, I was a bit confused when reading the abstract (though I could still guess to some extent). It would be helpful if the authors could elaborate more on this.***
> >
> > Thank you for your feedback. We will clarify these terms earlier in the abstract to prevent confusion. Generally, "semantic tokens" refer to tokens obtained through the quantization process of Self-Supervised Learning (SSL) models, whereas "compression tokens" (or codecs) involve the use of encoder-decoder architectures with Residual Vector Quantization (RVQ), which are trained to reconstruct the original audio accurately. While we acknowledge that these terms may not perfectly capture the categories, they are widely used in the literature [1,2], and we use them to maintain consistency with established terminology.
> > - [1]: Borsos, Zalán, et al. "Audiolm: a language modeling approach to audio generation." IEEE/ACM transactions on audio, speech, and language processing 31 (2023): 2523-2533.
> > - [2]: Zhang, Xin, et al. "Speechtokenizer: Unified speech tokenizer for speech large language models." arXiv preprint arXiv:2308.16692 (2023).
> >
> > >***I believe there have been several studies on hybrid tokens this year, but only SpeechTokenizer is included. This type of token is a very active area of research. Is there a chance of providing more results in this category?***
> >
> > We would like to note that at the time of submitting this benchmark, many new tokenizers had not yet been released or officially accepted at a conference. However, we are actively enhancing our benchmark by integrating more publicly available audio tokenizers, including Mimi, SQ-Codec (for scalar quantization), and WavTokenizer (for single-layer quantization). This ensures that we include a diverse range of tokenizer categories to provide a comprehensive evaluation.
> >
> > >***At the beginning of Section 4, there is an error in the Appendix index. Please fix it.***
> >
> > Thank you for pointing that out. We will address this in the next version.
> >
> > >***Which continuous SSL models are used exactly in Tables 2, 3, and 4?***
> >
> > We use the same configuration as the MP3S[1] benchmark for the continuous baselines. MP3S found that representations from the last layer of a model might not always be optimal. To address this, they utilized representations from all hidden layers of the pretrained model. These hidden states are then weighted and summed to create a representation for the decoder. Consistently, we employ Weighted WavLM for speech-related tasks, Weighted MERT for music tasks, and Weighted CNN14 for general audio tasks.
> > - [1]: Zaiem, Salah, et al. "Speech self-supervised representation benchmarking: Are we doing it right?." arXiv preprint arXiv:2306.00452 (2023).
> >
> > >***In ASR, the authors include two low-resource languages. I wonder if there are any specific reasons for choosing them. The evaluation results on these two languages are very poor across the models, and I think that instead of directly comparing them at this level, using phoneme recognition could lower the difficulty of the task and provide more comparable results.***
> >
> > We selected them based on the MP3S[1] benchmark We chose these languages because they are challenging tasks for low-resource languages, which are often not included in the training data for many tokenizers.

---

> > > ### Author Response · Authors · 2024-11-25
> > >
> > > ## Questions:
> > >
> > > >***For speech generation tasks, the authors use automatic MOS prediction models to evaluate the quality of generated speech. I understand that this is mainly because MOS involves human effort, and it is infeasible to use only the original MOS in a benchmark. However, as far as I know, the correlation between automatic MOS models and human ratings is still limited. Could you provide the correlation of these results with human ratings?***
> > >
> > > We understand and acknowledge the limitations of automated MOS estimation models, and we recognize that even human evaluations can be subjective and may vary significantly across different groups of evaluators or studies. To address this, we employed the UTMOS system, which is one of the highest-performing tools available. It achieved first place in 10 out of 16 metrics at the VoiceMOS Challenge[3]. Specifically, UTMOS reported a Spearman R correlation coefficient of 0.897 at the utterance level and 0.936 at the system level for in-distribution data, with 0.893 and 0.988, respectively, for out-of-distribution data.
> > >
> > > Our study uses UTMOS as a consistent metric for comparing standardized model architectures across different tokenizers and settings, rather than comparing it directly to MOS results from human studies reported elsewhere. This standardized approach is also adopted in other similar benchmarks such as ESPnet-Codec[2] and Codec-SUPERB[1]. Using a subjective metric like MOS could limit the scalability of our benchmark to include newly released tokenizers, as it would require ongoing human evaluations. Additionally, it may not be feasible to maintain consistency in human annotator groups over time, which could affect the fairness and reproducibility of the evaluations when new tokenizers are added to the benchmark.
> > > - [1] Wu, Haibin, et al. "Codec-SUPERB@ SLT 2024: A lightweight benchmark for neural audio codec models." arXiv preprint arXiv:2409.14085 (2024).
> > > - [2] Shi, Jiatong, et al. "ESPnet-Codec: Comprehensive Training and Evaluation of Neural Codecs for Audio, Music, and Speech." arXiv preprint arXiv:2409.15897 (2024).
> > > - [3] Huang, Wen-Chin, et al. "The VoiceMOS Challenge 2024: Beyond Speech Quality Prediction." arXiv preprint arXiv:2409.07001 (2024).
> > >
> > > >***In Tables 5 and 6, the discrete SSL parts are all replaced with CNN14 and MERT. Is there a chance to add results on the discrete HuBERT and WavLM? This could help test the generalizability from speech to audio and music.***
> > >
> > > HuBERT and WavLM are not trained on music or general audio data, making it potentially unfair to compare them directly with other tokenizers that have been trained on extensive music and audio datasets. That's why we utilize pretrained SSL models that are specifically trained for each domain to ensure fair comparisons. However, it could be interesting to explore how models like Hubert or WavLM, which are originally designed for the speech domain, perform in other areas such as music and sound. Our initial experiments in this direction have not shown promising results.

---

> ### Comment · Reviewer_XCwv · 2024-11-27
>
> I would like to clarify that several implementations are publicly available before the ICLR submission deadline. Below, I list some neural codec models with accessible checkpoints:
>
> 1. WavTokenizer (checkpoint released on September 9, 2024, on GitHub)
> 2. RepCodec (last commit on July 12, 2024, on GitHub)
> 3. FACodec (last commit on March 13, 2024, on Hugging Face)
> 4. SemanticCodec (last commit on August 25, 2024, on GitHub)
> 5. BigCodec (checkpoint released on September 4, 2024, on GitHub)
>
> (I do not include the links to avoid potential violations of ICLR’s anonymity guidelines.)
>
> Therefore, I believe that including only SpeechTokenizer limits the scope of the experiments.

---

### Official Review · Reviewer_MprN · 2024-11-04

**Soundness:** 2
**Presentation:** 3
**Contribution:** 2
**Rating:** 3
**Confidence:** 5

**Summary:**

This article provides a benchmark of discrete tokens of speech and audio on comprehension and generation tasks. This paper divides tokens into three categories, namely semantic, compression, and hybrid. Through a large number of experiments, the paper gives some conclusions.

**Strengths:**

1. The paper conducted plenty of experiments to evaluate the understanding and generation tasks of different types of speech tokens.
2. The presentation is clear and the results are easy to follow.

**Weaknesses:**

1. Training data of different types of tokens is different, in addition, the decoders(Vocoders) of semantic tokens are only trained on Librispeech 960. As a result, the horizontal comparison is not particularly fair, especially for music and sound.
2. The amount of experiments in the paper is large, but there is no obvious insight. Most of the conclusions are already common sense, such as semantic tokens are good at content and compression tokens are good at paralinguistics.
3. The experimental design and data of the paper are basically the same as SUPERB, Codec-SUPERB, and SUPERB-SG, with limited novelty.
4. I don't understand why in the author's experiment, the high bit rate is not as good as the medium one. I suspect there is some kind of implementation or hyperparameter setting problem.

**Questions:**

1. What is the continuous baseline? it doesn't seem clear.
2. Why are the SI and SV performance of SSL baseline best? It feels a little weird because they are semantic features.

---

> ### Author Response · Authors · 2024-11-25
>
> ## Weaknesses
>
> We thank the reviewer for their time and valuable feedback. We appreciate the constructive suggestions and are glad to receive insightful comments on our work. Following your feedback, we will update our paper accordingly.
>
> >***Training data of different types of tokens is different, in addition, the decoders(Vocoders) of semantic tokens are only trained on Librispeech 960. As a result, the horizontal comparison is not particularly fair, especially for music and sound.***
>
> We appreciate the concern you've raised regarding the fairness of comparisons due to differences in training data and the specific conditions under which various tokenizers and their decoders are trained. Indeed, achieving a fair comparison poses a challenge and can be approached in two distinct ways.
> The first approach involves standardizing the conditions under which each tokenizer is evaluated. This would mean retraining the architecture of these tokenizers in a controlled setting using the same dataset, bitrate, and sampling rate. This method can provide valuable insights into which architecture is inherently more effective, although it might deviate from the optimal settings and hyperparameters originally recommended by the authors, potentially affecting the performance as reported in their studies.
> The second approach, which we currently adopt, is to utilize the pretrained models as developed by the original authors. This method aims to assess how well each tokenizer performs as a feature extractor in its best possible configuration, as intended by its authors. This is in line with practices from other benchmarks like Superb, where not all SSL models are trained on the same data or share identical settings, yet a consistent setting for training downstream tasks is used to ensure fair comparisons.
> To address discrepancies, particularly concerning bitrate, we provide three different settings, low, medium, and high,  for each tokenizer and compare the performance within each category. Additionally, similar to a good practice suggested for assessing self-supervised learning models proposed in [1], we are considering two different downstream architectures, which makes our results less noisy than what is often done. This strategy helps to mitigate the impact of varying conditions and allows for a more equitable assessment of each model's capabilities.
>   -  [1]: Zaiem, Salah, et al. "Speech self-supervised representation benchmarking: Are we doing it right?." arXiv preprint arXiv:2306.00452 (2023)

---

> > ### Author Response · Authors · 2024-11-25
> >
> > ## Weaknesses
> > > ***The amount of experiments in the paper is large, but there is no obvious insight. Most of the conclusions are already common sense...The experimental design and data of the paper are basically the same as SUPERB, Codec-SUPERB, and SUPERB-SG, with limited novelty.***
> >
> > As noted in the related works section, both Codec-SUPERB and ESPnet-Codec focus on evaluating the quality of resynthesized sound, utilizing compression and hybrid tokenizers. They have conducted comprehensive study using pretrained models for speech, speaker, and emotion recognition to determine how much information the ***resynthesized audio*** retains. However, these studies have not directly addressed the use of tokenized inputs for training ***downstream tasks*** or analyzed the role of ***semantic tokens***. While  ESPnet-Codec addresses some downstream tasks, it does not offer as comprehensive a coverage as our DASB benchmark, and it employs more complex architectures. In both Codec-SUPERB and ESPnet-Codec, the good results may disproportionately reflect the capabilities of the decoders rather than the tokens themselves.
> > Our DASB benchmark is designed specifically to assess the efficacy of audio tokens using relatively simple downstream architectures. This approach ensures the focus remains on the tokens' quality. By avoiding overly complex architectures, we prevent them from compensating for less effective tokens, thereby maintaining an equitable basis for comparison. This strategy aligns with established benchmarks like SUPERB[3], USB[1], and MP3S[2], which are all designed to evaluate the effectiveness of speech self-supervised models. Our intent is thus not to claim state-of-the-art results for specific downstream tasks but rather to explore how each tokenizer varies in its capacity to preserve specific information and identify its limitations.
> >
> > - [1]: Wang, Yidong, et al. "Usb: A unified semi-supervised learning benchmark for classification." Advances in Neural Information Processing Systems 35 (2022): 3938-3961.
> > - [2]: Zaiem, Salah, et al. "Speech self-supervised representation benchmarking: Are we doing it right?." arXiv preprint arXiv:2306.00452 (2023).
> > -  [3]: Yang, Shu-wen, et al. "Superb: Speech processing universal performance benchmark." arXiv preprint arXiv:2105.01051 (2021).
> >
> > >***I don't understand why in the author's experiment, the high bit rate is not as good as the medium one. I suspect there is some kind of implementation or hyperparameter setting problem.***
> >
> > Recent studies have shown that higher bit rates do not always lead to better performance on downstream tasks. For example, Moshi[1] mentioned a counterintuitive trend: the benefits of lower bit rates become more significant as we reduce the bit rate. Additionally, the VoxtLM [2] study reported that using 200 centroids achieved a Character Error Rate (CER) of 3.5 for Text-to-Speech (TTS), whereas increasing the number of centroids to 1000 resulted in a higher CER of 6.1. This suggests that a higher bit rate does not necessarily lead to better outcomes. A higher bit rate in an audio tokenizer can sometimes impair performance by generating an excessive number of discrete tokens. This over-tokenization can introduce complexity and redundancy, which may obstruct the model's capacity to meaningfully interpret semantic information from the audio, despite potentially higher quality in audio reconstruction.
> >
> > - [1]: Défossez, Alexandre, et al. "Moshi: a speech-text foundation model for real-time dialogue." arXiv preprint arXiv:2410.00037 (2024).
> > - [2]: Maiti, Soumi, et al. "VoxtLM: Unified Decoder-Only Models for Consolidating Speech Recognition, Synthesis and Speech, Text Continuation Tasks." ICASSP 2024-2024 IEEE International Conference on Acoustics, Speech and Signal Processing (ICASSP). IEEE, 2024.

---

> > > ### Author Response · Authors · 2024-11-25
> > >
> > > ## Questions:
> > >
> > > >***What is the continuous baseline? it doesn't seem clear.***
> > >
> > > We use the same configuration as the MP3S[1] benchmark for the continuous baselines. MP3S found that representations from the last layer of a model might not always be optimal. To address this, they utilized representations from all hidden layers of the pretrained model. These hidden states are then weighted and summed to create a representation for the decoder. Consistently, we employ Weighted WavLM for speech-related tasks, Weighted MERT for music tasks, and Weighted CNN14 for general audio tasks.
> > > - [1]: Zaiem, Salah, et al. "Speech self-supervised representation benchmarking: Are we doing it right?." arXiv preprint arXiv:2306.00452 (2023).
> > >
> > > >***Why are the SI and SV performance of SSL baseline best? It feels a little weird because they are semantic features.***
> > >
> > >  The continuous baseline indeed performs best, particularly in SI and SV tasks. This can be attributed to its ability to retain detailed speaker-specific information, which is often lost during quantization. Among the discrete tokenizers, those based on compression outperform those based on semantic methods. This is because compression-based models are specifically trained to reconstruct audio accurately, preserving fine-grained details that are crucial for speaker identification. In contrast, semantic tokenizers focus on capturing higher-level content and abstract features, which might not include all the nuances necessary for precise speaker recognition.

---

> > > > ### Comment · Reviewer_MprN · 2024-11-28
> > > > **Re**
> > > >
> > > > Thank you for your feedback. However, I still find it difficult to fully agree with the contributions presented in the paper. The insights provided appear to be limited, and the incremental nature of the benchmark design (especially in comparison to the SUPERB series) does not offer a significant advancement in the field. Furthermore, the incomplete evaluation within the audio domain, as pointed out by Reviewer 8g5d, raises concerns regarding the thoroughness of the study. Given these points, I keep my initial rating.

---

### Official Review · Reviewer_wdqV · 2024-11-08

**Soundness:** 3
**Presentation:** 3
**Contribution:** 3
**Rating:** 6
**Confidence:** 4

**Summary:**

- The authors offer a benchmark for evaluating the quality of neural audio codecs and assessing their impact on various downstream tasks.
- These downstream tasks encompass several standard audio processing applications, including speech, sound, and music.

**Strengths:**

- This is an open-source project dedicated to enhancing the quality of neural audio codecs, which are becoming increasingly valuable to the community as their importance grows.
- It provides an exciting opportunity to benchmark performance across a variety of downstream tasks—not only for speech but also for broader audio and music applications

**Weaknesses:**

- I've noticed a surge in similar open-source projects released recently—some of which were published after the ICLR submission deadline, highlighting the relevance of this work. However, I have some concerns regarding its future adoption. To encourage broader usage, it might be beneficial to emphasize the unique advantages of this repository and clarify why researchers would prefer it over others. For instance, a comparison table could be added to showcase its strengths in contrast to similar projects.

Examples of recent related works include: [1] Wu, Haibin, et al. "Codec-SUPERB@ SLT 2024: A lightweight benchmark for neural audio codec models." arXiv preprint arXiv:2409.14085 (2024). [2] Shi, Jiatong, et al. "ESPnet-Codec: Comprehensive Training and Evaluation of Neural Codecs for Audio, Music, and Speech." arXiv preprint arXiv:2409.15897 (2024).

- Inference time may be quite lengthy, potentially taking up to 8 hours for tasks like ASR. Providing a smaller development set for quick experiments could be a valuable addition.

**Questions:**

- In addition to the GitHub repository, do the authors plan to provide a leaderboard for tracking and comparing results?
- Would it be possible to include a comparison of computational costs as well?
- Do the authors have plans to add support for higher sample rates in future comparisons?
- Since a codec might perform well on one downstream task but poorly on another, what approach would you recommend to achieve balanced results for selecting a universally effective codec?

---

> ### Author Response · Authors · 2024-11-25
>
> ## Weaknesses
>
> We thank the reviewer for their time and valuable feedback. We appreciate the constructive suggestions and are glad to receive insightful comments on our work. Following your feedback, we will update our paper accordingly.
>
> >***I've noticed a surge in similar open-source projects released recently... However, I have some concerns regarding its future adoption. To encourage broader usage, it might be beneficial to emphasize the unique advantages of this repository and clarify why researchers would prefer it over others...***
>
> As noted in the related works section, both Codec-SUPERB and ESPnet-Codec focus on evaluating the quality of resynthesized sound, utilizing compression and hybrid tokenizers. They have conducted comprehensive study using pretrained models for speech, speaker, and emotion recognition to determine how much information the **resynthesized audio** retains. However, these studies have not directly addressed the use of tokenized inputs for training **downstream tasks** or analyzed the role of **semantic tokens**. While  ESPnet-Codec addresses some downstream tasks, it does not offer as comprehensive a coverage as our DASB benchmark, and it employs more complex architectures. In both Codec-SUPERB and ESPnet-Codec, the good results may disproportionately reflect the capabilities of the decoders rather than the tokens themselves.
> Our DASB benchmark is designed specifically to assess the efficacy of audio tokens using relatively simple downstream architectures. This approach ensures the focus remains on the tokens' quality. By avoiding overly complex architectures, we prevent them from compensating for less effective tokens, thereby maintaining an equitable basis for comparison. This strategy aligns with established benchmarks like SUPERB[3], USB[1], and MP3S[2], which are all designed to evaluate the effectiveness of speech self-supervised models. Our intent is thus not to claim state-of-the-art results for specific downstream tasks but rather to provide a fair comparison of different tokenizers.
>
> - [1]: Wang, Yidong, et al. "Usb: A unified semi-supervised learning benchmark for classification." Advances in Neural Information Processing Systems 35 (2022): 3938-3961.
> - [2]: Zaiem, Salah, et al. "Speech self-supervised representation benchmarking: Are we doing it right?." arXiv preprint arXiv:2306.00452 (2023).
> -  [3]: Yang, Shu-wen, et al. "Superb: Speech processing universal performance benchmark." arXiv preprint arXiv:2105.01051 (2021).
>
> ## Questions:
> > ***In addition to the GitHub repository, do the authors plan to provide a leaderboard for tracking and comparing results?***
>
> We will add a reference to the DASB website for the camera-ready version for anonymity reasons, which includes a tutorial for adding and evaluating new tokenizers. The website also features a leaderboard for discrete units. Furthermore, we are actively working to expand the benchmark by incorporating more publicly available audio tokenizers. We plan to introduce a challenge to encourage researchers to evaluate their tokenizers using DASB. Built on the SpeechBrain toolkit, which is actively maintained, DASB ensures that we can regularly update the benchmark with the latest advancements.
>
> > ***Would it be possible to include a comparison of computational costs as well?***
>
>  We've already analyzed computational costs by measuring the time and memory required to process a 16-second utterance for the encoders and decoders of each tokenizer, as shown in Figure 2. We also detail the number of parameters, sampling rates, and bitrates for each tokenizer in Table 1.
> To further this analysis, we can add additional metrics such as token rate, frame rate, and MACs to offer a deeper view on the computational demands of each system.
>
> > ***Do the authors have plans to add support for higher sample rates in future comparisons?***
>
> We have utilized the original and suggested sampling rates for each tokenizer, employing their pretrained checkpoints. For DAC, we opted for a 24kHz sampling rate instead of 44kHz, aligning with what is commonly used in the literature. However, should there be new tokenizers in the future that perform optimally at higher sampling rates, we are open to including those as well. It's worth noting that the current trend, especially in the speech domain, typically centers around 16kHz, though sometimes it extends to 24kHz.
>
> > ***Since a codec might perform well on one downstream task but poorly on another, what approach would you recommend to achieve balanced results for selecting a universally effective codec?***
>
> This is an essential question and the primary motivation behind our benchmark. With DASB, we aim to explore how each tokenizer varies in its capacity to preserve specific information and identify its limitations. This analysis will serve as a valuable guide for future research, helping to develop a tokenizer that can effectively perform across a diverse range of tasks universally.

---

### Author Response · Authors · 2024-11-25

We thank the reviewers for their feedback.  We are glad to see that all reviewers (8g5d, XCwv, MprN,wdqV ) unanimously recognized the diversity and comprehensiveness of our benchmark. As DASB is a community-driven project, we are committed to incorporating feedback from both reviewers and the broader community. We have posted detailed responses to each reviewer's comments

---

### Meta-Review · Area_Chair_FGcq · 2024-12-13

**Metareview:**

In this work, the authors presents DASB, a benchmark for evaluating how well architectures generating "discrete" audio tokens perform on various speech and audio tasks, including discriminative tasks (e.g., ASR, classification) and generative ones (e.g., TTS, speech enhancement). The benchmark focuses on three token types: semantic-based, compression-based, and hybrid. It examines several models across these categories, analyzing how bitrates impact compression-based token quality.

Four independent reviewers assessed the work and despite the different small nuances among the overall the final reccomandations, the work seems not ready for publication. In fact, even after a good discussion phase between authors and reviewers, two of the reviewers kept their negative opinion suggesting to reject the work. Futhermore, although the remaning two reviewers were for a weak acceptance, one of them has however stressed out that "including only SpeechTokenizer limits the scope of the experiments."

Despite the interesting issue on Audio Representation Learning addressed in the proposed work, namely evaluating the quality of representations, in this specific case given by discrete tokens. Several major concerns were not resolved after the discussion period, namely: (i) The incremental nature of the benchmark design (especially in comparison to the SUPERB series) does not offer a significant advancement in the field. (ii) The evaluation is  incomplete within the audio domain,  which raises concerns regarding the thoroughness of the study, (iii) Some of the authors' claim not fully supported by facts, e.g., referencing to [1] to justify using 1000 clusters.

Finally, the following three questions have apparently been left unswered: (a) What unique insights does DASB provide beyond existing frameworks? (b) Why couldn't these tokens be evaluated using existing benchmarks? (c) How the DASB methodological choices compare to evaluation approaches in representation learnig (e.g. linear probing)?

**Additional Comments On Reviewer Discussion:**

The discussion between authors and reviewers was good. Nonetheless, the key concerns were not address by the authors. The following points remained unresolved: (i) The incremental nature of the benchmark design (especially in comparison to the SUPERB series) does not offer a significant advancement in the field. (ii) The evaluation is  incomplete within the audio domain,  which raises concerns regarding the thoroughness of the study, (iii) Some of the authors' claim not fully supported by facts, e.g., referencing to [1] to justify using 1000 clusters.

---

### Decision · Program_Chairs · 2025-01-22

Reject